

# Record breaking achievements by spiders and the scientists who study them

Stefano Mammola[1,2], Peter Michalik[3], Eileen A. Hebets[4] and Marco Isaia[1,2]

[1] Department of Life Sciences and Systems Biology, University of Turin, Torino, Italy
[2] IUCN SSC Spider and Scorpion Specialist Group, Torino, Italy
[3] Zoologisches Institut und Museum, Ernst-Moritz-Arndt Universität Greifswald, Greifswald, Germany
[4] School of Biological Sciences, University of Nebraska–Lincoln, Lincoln, NE, USA

## ABSTRACT

Organismal biology has been steadily losing fashion in both formal education and scientific research. Simultaneous with this is an observable decrease in the connection between humans, their environment, and the organisms with which they share the planet. Nonetheless, we propose that organismal biology can facilitate scientific observation, discovery, research, and engagement, especially when the organisms of focus are ubiquitous and charismatic animals such as spiders. Despite being often feared, spiders are mysterious and intriguing, offering a useful foundation for the effective teaching and learning of scientific concepts and processes. In order to provide an entryway for teachers and students—as well as scientists themselves—into the biology of spiders, we compiled a list of 99 record breaking achievements by spiders (the "Spider World Records"). We chose a world-record style format, as this is known to be an effective way to intrigue readers of all ages. We highlighted, for example, the largest and smallest spiders, the largest prey eaten, the fastest runners, the highest fliers, the species with the longest sperm, the most venomous species, and many more. We hope that our compilation will inspire science educators to embrace the biology of spiders as a resource that engages students in science learning. By making these achievements accessible to non-arachnologists and arachnologists alike, we suggest that they could be used: (i) by educators to draw in students for science education, (ii) to highlight gaps in current organismal knowledge, and (iii) to suggest novel avenues for future research efforts. Our contribution is not meant to be comprehensive, but aims to raise public awareness on spiders, while also providing an initial database of their record breaking achievements.

## INTRODUCTION

Organismal biology, or the study of the structure, function, ecology and evolution of organisms, is critical for understanding the fundamental questions in ecology, evolutionary biology, neurobiology, and more. In other words, organismal biology is essential for science—for its practice, its teaching, and its learning (*Schwenk et al., 2009*).

Corresponding authors
Stefano Mammola,
stefano.mammola@unito.it
Marco Isaia, marco.isaia@unito.it

The intensive study and detailed understanding of specific organisms enables research programs that can address important and timely questions and topics, such as climate change, disease transmission, pest management, and biomaterial engineering (*Maher, 2009*; *Alfred & Baldwin, 2015*). The natural world around us harbors surprises that even the most educated and creative minds could not fashion de novo (*Bonabeau, Dorigo & Theraulaz, 2000*; *Sarkar, Phaneendra & Chakrabarti, 2008*; *Place, Evans & Stevens, 2009*; *Grzybowski & Huck, 2016*). Thus, the study of organisms allows scientists and non-scientists alike to travel outside the limits of their own imagination.

Unfortunately, as a species, *Homo sapiens* is losing its collective knowledge, understanding, and appreciation of the organisms with which it shares the planet. There exists a growing trend for youth and adults alike to be increasingly physically inactive and, associated with this, to spend less and less time outdoors (*Guthold et al., 2010*; *Hallal et al., 2012*; *Schaefer et al., 2014*; *Tremblay et al., 2014*). Simultaneously, as science funding becomes harder and harder to acquire, basic natural history information and organism-based studies are more difficult to not only justify, but also to publish (*Greene, 2005*; *Middendorf & Pohlad, 2014*; *Tewksbury et al., 2014*; *LoPresti et al., 2016*). Additionally, in higher education there has been an increasing emphasis on pedagogical tools and practices that focuses on learning objectives, associated with broad concepts and critical thinking, with less focus on skills of observation and foundational facts associated with organismal biology (*McLaughlin & Metz, 2016*; *Fleischner et al., 2017*). The result is that it is more and more difficult to expose teachers, learners of science, and scientists themselves to the incredible wealth of facts, wonders, and curiosities offered by organismal biology—see, e.g., the numerous examples in *Carwardine (2008)*.

Despite the movement away from organismal biology among the general public, teachers and students of science, as well as among many scientists, human curiosity and intrigue persists. This curiosity and intrigue is best demonstrated by the purity with which it is observed in our youth. Some of the first words that children learn or noises children make are animal-specific—e.g., the multiple "first words" books for babies and toddlers (*Priddy, 2004*; *Machell, 2005*). Similarly, animal-related stories are common among early reading children's books, presumably because they can attract and retain a child's interest and attention. Even among adults, animals remain a useful tool for attracting attention and making connections among diverse societies, as evidenced by the numerous viral videos focused on cats, dogs, and other animals. Following from these observations of human interest in animals, we contend that organismal biology, especially the biology of particularly charismatic organisms, can still be an extraordinarily useful tool for engaging people of all ages in science-related teaching and learning and importantly can remain a source of inspiration for innovate, ground-breaking scientific studies.

Spiders and arachnids in general, are animals that can simultaneously instill both terror and intrigue. Their charismatic nature makes it extraordinarily easy to attract even the most bio-phobic individual into arachnid-based discussions and activities. Arachnids tend to be either loved or feared (and "hated"), with few people feeling ambivalence toward them (*Hillyard, 1994*; *Mulkens, de Jong & Merckelbach, 1996*; *Woody, McLean &*

*Klassen, 2005*; *Rinck & Becker, 2007*; *Knight, 2008*). Even a fear of spiders, however, can be harnessed toward the goal of enhancing science teaching and learning, because they are able to evoke such strong reactions. For example, arachnophobic individuals in particular demonstrate enhanced recall to spider-relevant information (*Smith-Janik & Teachman, 2008*).

In addition to their charismatic nature, spiders are widespread and abundant, making them familiar and readily accessible to people everywhere. Compared to most organisms, they are understudied, thus providing opportunities for scientific discovery that could empower scientists and non-scientists alike with prospects of personal scientific contributions. They are also suitable model organisms for laboratory and field experimentation, making it easy to facilitate hands-on science. Perhaps most importantly, however, is the fact that spider ecology and evolution is fertile ground for teaching a breadth of science, technology, engineering, and mathematic (STEM) concepts.

For example, spider silk can be used to explore topics ranging from evolution of form and function, to biomaterial engineering, to the physical properties of protein fibers (*Hinman, Jones & Lewis, 2000*; *Heim, Keerl & Scheibel, 2009*). Knowledge of spider natural history and habitat use can inform pest management practices (*Nyffeler & Benz, 1987*; *Marc & Canard, 1997*), and biodiversity conservation efforts (*Cardoso et al., 2004*). Spider sensory and locomotory systems can inspire technological innovation (*King, 2013*; *Kang et al., 2014*) and spider venom can inspire medical and pharmaceutical innovations (*Bode, Sachs & Franz, 2001*; *Saez et al., 2010*; *King & Hardy, 2013*). In essence, we contend that spider biology can be used as a foundation for teaching a range of topics and subjects at any level of education (K-12 or higher education). However, to facilitate the implementation of spider biology as a resource for teaching, learning, and research inspiration, the scientific background information needs to be accurate and accessible— and preferably published in a clear and enjoyable way (*Sand-Jensen, 2007*; *Heard, 2014*).

Toward our goal, we compiled a database of record breaking spider achievements. In presenting our database, we take advantage of the reality that humans often tend to think in extremes. Indeed, for people of all ages, the entire range of superlatives exerts a powerful spell on human curiosity. Scientists are no exception, as they are similarly attracted by formidable species and record breaking biological discoveries (*Watson & Walker, 2004*; *Edwards et al., 2005*; *Glaw et al., 2012*; *Sendra & Reboleira, 2012*; *Wilson et al., 2012*; *Andersen et al., 2016*; *Klug et al., 2015*; *McClain et al., 2015*). Thus, we present our findings in a world-record style format, as this is known to be an effective way to engaging youth and adults alike.

Numerous spider-related world records have already been claimed in peer-review scientific papers (*Jäger, 2001*; *Kuntner & Coddington, 2009*; *Agnarsson, Kuntner & Blackledge, 2010*; *Lepore et al., 2012*; *Smithers & Whitehouse, 2016*). Officially, spiders hold 44 Guinness World Records (hereinafter GWR) related to their biology (see *GWR, 2017*; full list in Supplemental information). Here, we explore the scientific literature to provide a broader overview of record breaking achievements by spiders (Table 1). We in no way intend this to be an exhaustive list, but more of a "highlight" that can provide an entryway into the biology of spiders. Our goal is to make these

**Table 1 General organization of the Spider World Records.**

| | | |
|---|---|---|
| I. Arachnology and arachnologists | | a. *First arachnologist in history* |
| | | b. *Most prolific arachnologist* |
| | | c. *First catalogue of spiders* |
| | | d. *Longest publication on spiders* |
| | | e. *First congress of arachnology* |
| | | f. *Most attendees at a congress of arachnology* |
| II. Paleontology | | a. *First described fossil* |
| | | b. *Oldest fossil spider* |
| | | c. *Oldest fossil spider in amber* |
| | | d. *Oldest recorded spider silk* |
| | | e. *Oldest web with entrapped prey* |
| | | f. *Oldest recorded predatory event* |
| | | g. *Oldest social spider* |
| | | h. *Largest fossil spider* |
| III. Taxonomy and Systematics | | a. *First spider(s) ever described in binomial nomenclature* |
| | | b. *First listed spider alphabetically* |
| | | c. *Last listed spider alphabetically* |
| | | d. *Longest scientific name* |
| | | e. *Shortest scientific name* |
| | | f. *Largest spider family* |
| | | g. *Smallest spider family* |
| | | h. *First entire genome sequenced* |
| | | i. *Most species named after celebrities within one genus* |
| IV. Anatomy | 1. Size | a. *Largest living spiders* |
| | | b. *Smallest adult female spider* |
| | | c. *Smallest adult male spider* |
| | | d. *Most extreme sexual size dimorphism* |
| | | e. *Most unusual sexual size dimorphism* |
| | 2. Body parts | a. *Highest number of eyes* |
| | | b. *Least number of eyes* |
| | | c. *Largest eyes* |
| | | d. *Longest relative chelicerae* |
| | | e. *Largest relative fangs* |
| | | f. *Longest relative walking legs* |
| | | g. *Most legs* |
| | | h. *Most spinnerets* |
| | | i. *Longest relative spinnerets* |
| | 3. Internal organs | a. *Largest central nervous system* |
| | | b. *Largest relative venom glands* |
| | | c. *Smallest relative venom glands* |

| Table 1 (continued). | | |
|---|---|---|
| V. Physiology | 1. Silk and webs | a. *Largest web (area)* |
| | | b. *Largest web (length)* |
| | | c. *Smallest web* |
| | | d. *Strongest silk* |
| | | e. *Strongest cocoon silk* |
| | 2. Venom | a. *Most venomous to humans* |
| | | b. *Least venomous* |
| | | c. *Most unusual venom* |
| | 3. Sensory organs | a. *Best diurnal eyesight* |
| | | b. *Best nocturnal eyesight* |
| | | c. *Best hearing* |
| | | d. *Most bioluminescent* |
| | 4. Biological cycle | a. *Longest life span* |
| | | b. *Shortest circadian rhythm* |
| | 5. Eggs and sperms | a. *Longest sperm* |
| | | b. *Highest number of eggs* |
| | | c. *Least number of eggs* |
| VI. Behavior | 1. Locomotion | a. *Best ballooners* |
| | | b. *Best sailors* |
| | | c. *Fastest spider* |
| | | d. *Fastest rotational movement* |
| | 2. Foraging | a. *Most creative hunting strategies* |
| | | b. *Fastest predatory strike* |
| | | c. *Largest invertebrate prey* |
| | | d. *Largest vertebrate prey* |
| | | e. *Strangest diet* |
| | | f. *Fussiest spider* |
| | 3. Reproduction | a. *Shortest Mating* |
| | | b. *Longest mating* |
| | | c. *Best date* |
| | | d. *Most elaborate courtship* |
| | | e. *Most complex song* |
| | | f. *Loudest spider* |
| | | g. *Best mother* |
| | | h. *Best father* |
| | 4. Lifestyle | a. *Most peaceful* |
| | | b. *Largest colony* |
| | | c. *Best thieves* |
| | | d. *Best camouflage* |
| | | e. *Longest time under water* |
| | | f. *Longest time under water in a nest* |

(Continued)

| Table 1 (continued). | | |
|---|---|---|
| VII. Ecology | 1. Habitat | a. *Highest altitude* |
| | | b. *Lowest altitude* |
| | | c. *Coldest place inhabited by spiders* |
| | | d. *Hottest place inhabited by spiders* |
| | | e. *Northernmost species* |
| | | f. *Southernmost species* |
| | | g. *Most diverse habitat* |
| | | h. *Least suitable habitat* |
| | | i. *Strangest habitat* |
| | 2. Conservation | a. *Rarest* |
| | | b. *Most endangered* |
| | | c. *Most wanted as pet* |
| VIII. Curiosities | | a. *The longest journey* |
| | | b. *Most delicious* |
| | | c. *Most eaten by humans* |
| | | d. *Most feared* |
| | | e. *Largest item of clothing woven from spider silk* |
| | | f. *Most iconic spider* |

achievements accessible to both non-arachnologists and arachnologists. We suggest that such a database can: (i) be used by educators to draw in students for science education, (ii) highlight gaps in current organismal knowledge, and (iii) suggest novel avenues for future research efforts.

We begin our synthesis of record breaking achievements with a brief introduction to spiders followed by a presentation of record breaking achievements organized into eight distinct categories (Table 1).

## A BRIEF INTRODUCTION TO SPIDERS

Spiders (Araneae) belong to the class Arachnida together with 10 other orders: scorpions (Scorpiones), harvestmen (Opiliones), pseudoscorpions (Pseudoscorpiones), windscorpions (Solifugae), mites and ticks ("Acari"), micro-whip scorpions (Palpigradi), hooded tickspiders (Ricinulei), tailless whipscorpions (Amblypygi), and shorttailed whipscorpions (Schizomida) and whipscorpions (Uropygi)—common names based on *Breene et al. (2003)*. All spiders are hypothesized to have descended from a common ancestor (i.e., they represent a monophyletic group; *Garrison et al., 2016*; *Wheeler et al., 2016*) and the group encompasses nearly 47,000 extant species, distributed among 4,072 genera and 112 families (*WSC, 2017*). They are considered to be one of the most successful groups of organism in terms of their long evolutionary history and diverse ecological impacts—they are distributed in virtually all terrestrial ecosystems and play a key role as generalist carnivorous predators (*Turnbull, 1973*; *Foelix, 2011*). Indeed, a recent study by *Nyffeler & Birkhofer (2017)* estimated that the global spider community consumes between 400 and 800 million tons of prey annually.

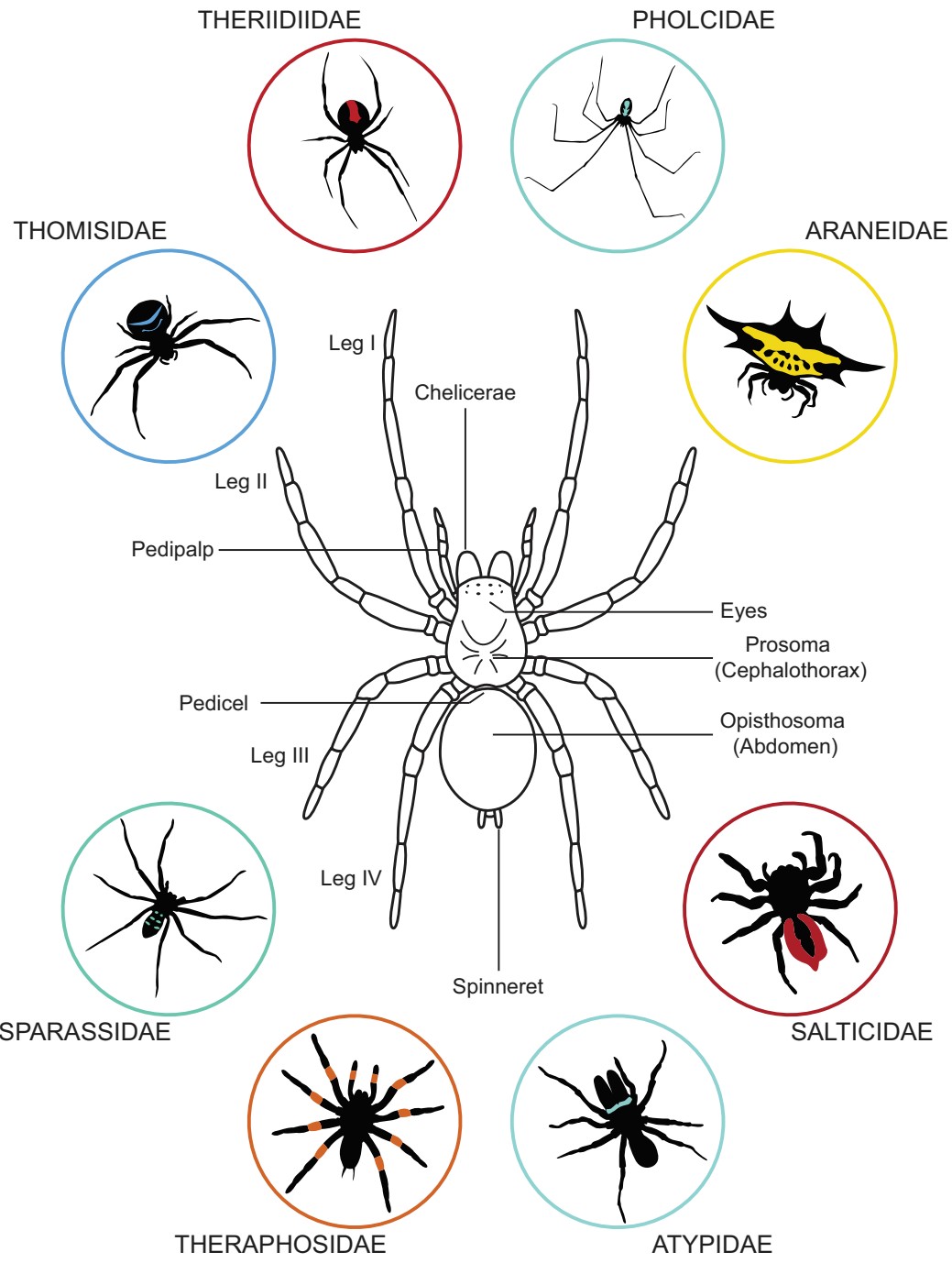

**Figure 1 General anatomy of a spider and variation in body forms.** Dorsal view of a spider showing its general organization and variation in its appearance exemplified by a few representative of the 112 known spider families.

The body of a spider is divided in two parts: (i) the prosoma (or cephalothorax) and (ii) the opisthosoma (or abdomen). These two body parts are joined by a narrow stalk called a pedicel (Fig. 1). The prosoma is relatively hard and carries six pairs of appendages: the chelicerae, the pedipalps, and four pair of walking legs. The chelicerae function in spider

feeding and venom injection takes place through their fangs. Posterior to the chelicerae are the pedipalps—the first pair of appendages behind the mouth. The pedipalps of adult males are modified into copulatory organs and facilitate the transfer of sperm to mature females. The four pairs of walking legs are posterior to the pedipalps. All walking legs originate from the prosoma, unlike the way they are sometimes portrayed in spider merchandise—e.g., attached to a single body part or inaccurately originating from the opisthosoma. In addition to the six pairs of appendages, the eyes are also located on the prosoma. Most spiders possess eight eyes, but in some species this number may be reduced or eyes may be entirely lacking. Though they do not have traditional ears, spiders can detect vibrations with slits in their cuticle (slit sensilla and lyriform organs) located on their walking legs. They can also detect airborne particle movement with long thin hairs located across their body.

The second body part—the opisthosoma—is soft, expandable, and shows high variation in shape and pattern among species (Fig. 1). The abdomen of spiders houses the respiratory system, the heart and most of the circulatory system, most of the digestive system, the excretory system, the silk producing system, and the reproductive system. In addition to these internal systems, the genital openings are located on the underside (i.e., ventral surface) and are barely visible in mature males and immatures. In females of most spiders, the genital opening can be covered by a hardened (i.e., sclerotized) structure, the epigyne. At the back end of the opisthosoma most spiders have their spinnerets, which are used for producing silk. Depending upon the species, a single spider can possess up to eight types of silk glands, each extruding a distinct type of silk. Silk is deployed in almost every aspect of a spider's life, from web construction to egg protection (*Foelix, 2011*).

## METHODS

We began the compilation of the Spider World Records by verifying all available biological records on spiders reported in the GWR official database (see Supplemental information). Wherever we observed discrepancies between the information found in the official GWR and that found in the scientific literature, we provide details in the relative record sections. A thorough search of the available literature was then conducted to track further documentations of extremes in spider biology. This included finding peer-reviewed articles by means of literature searches engines (Google Scholar, Scopus, Web of Sciences) but also personal communications with arachnologists and other scientists conducting research on the topics under evaluation (i.e., expert-based opinion). Most records related to taxonomy were compiled exploring the online catalog of spiders (*WSC, 2017*), including updated species counts and all literature on spider taxonomy from 1757 to date.

## SPIDER WORLD RECORDS

### Arachnology and Arachnologists

(a) *First arachnologist in history—Carl Alexander Clerck (1709–1865).* Although reports about spiders can be found in very old writings such as those of Aristotle and Pliny, according to *Bonnet (1955)* the father of the modern arachnology was Carl Alexander Clerck, author of the first book on spiders using the binomial system of nomenclature,

*Svenska Spindlar* (*Clerck, 1757*). His book was published only one year before the seminal "*Systema Naturae*" of Carl von Linné (*Linnaeus, 1758*), which marks the beginning of the binomial system of nomenclature. In order to consider Clerck's spider descriptions valid under the system of zoological nomenclature, his work is deemed to be published on 1 January 1758, which is regulated in the International Code of Zoological Nomenclature (Article 3.1; *ICZN, 1999*) (Fig. 2A).

(b) *Most prolific arachnologist—Eugène Louis Simon (1848–1924)*. In terms of publications, the most prolific arachnologist was the French naturalist Eugène Louis Simon. Over his life, he authored more than 270 spider-related scientific works, and he described (or revised the status) of 5,633 species—although some of them were later synonymized or considered *nomen dubia* (*WSC, 2017*) (Fig. 2B).

(c) *First catalogue of spiders—1942*. Carl Friedrich Roewer (1881–1963) own the record for publishing the first catalogue of spiders, i.e., the first volume of "*Katalog der Araneae von 1758 bis 1940*," published in 1942 (*Roewer, 1942*; see also *Roewer, 1955*). It included the list of spider species, synonyms and references published from 1758 to 1940. This remarkable publication provided the baseline, together with the competing catalog of *Bonnet (1955*, *1956*, *1957*, *1958*, *1959)* for further implementations (*Brignoli, 1983*; *Platnick, 1989*, *1993*, *1998*), up to the complete online taxonomic catalogues of spiders developed in the last decades (*Platnick, 2000–2014*; *WSC, 2017*; see also *World Spider Catalog Archive, 2014–2017*).

(d) *Longest publication on spiders—Bibliographia Araneorum*. With 6,481 pages, the longest publication on spiders is the *Bibliographia Araneorum* (*Bonnet, 1955*, *1956*, *1957*, *1958*, *1959*), representing the culmination of 40 years of work of the French arachnologist Pierre Bonnet (1897–1990).

(e) *First congress of Arachnology—Germany, 1960*. The first scientific arachnological meeting was held at the University of Bonn (Germany) in 1960. It was organized by Ernst Kullmann (1931–1996). According to the congress photo, at least 18 arachnologists attended this meeting (*Kraus, 1999*) (Fig. 2C).

(f) *Most attendees at a congress of arachnology—365*. With 365 participants, the 20th International Congress of Arachnology (2–9 July 2016, Golden, Colorado, USA) was the largest arachnological congress ever held. It was organized by Paula Cushing (Denver, USA) (Fig. 2D).

## Paleontology

(a) *First described fossil—An amber spider*. According to *Selden & Penney (2010)*, the earliest illustration of a fossil spider (an unidentified amber spider) can be found in *Kundmann (1737*: plate XII, Fig. 13*)*.

(b) *Oldest fossil spider—~300 Myr ago*. The oldest known true spiders date back to the Carboniferous age, around 300 Myr ago. Most likely, specimens of *Palaeothele montceauensis* (Selden) (Mesothelae) are the earliest described fossil species, from the Upper Carboniferous (Stephanian) of Montceau-Ies-Mines, France (*Selden, 1996*).

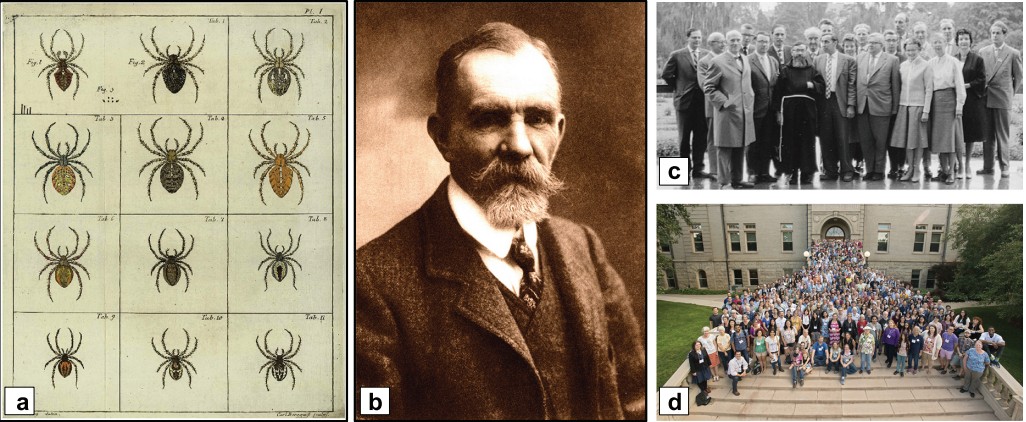

**Figure 2 Taxonomy, arachnology and arachnologists.** (A) Original illustrations of some of the first spiders described in binomial nomenclature (Modified from *Clerck (1757)*); (B) Eugène Louis Simon (1848–1924), the most prolific arachnologist in history (Photo credit: en.wikipedia.org); (C) The first Congress of Arachnology in history at the University of Bonn (Germany) in 1960 (Modified from *Kraus, 1999*); (d) The largest congress of Arachnology (2–9 July 2016, Golden, Colorado, USA) (photo credit: Paula Cushing—Congress Organizing committee).               

(c) *Oldest fossil spider in amber—125–135 Myr ago.* The oldest described amber spider (125–135 Myr ago) is an undetermined Linyphiinae, preserved in a small piece of Lebanese amber (*Penney & Selden, 2002*; *GWR, 2017*). This fossil specimen is also the oldest linyphiid spider known to date.

(d) *Oldest recorded spider silk—~140 Myr ago.* Although spiders are known to produce silk since the Mid-Devonian (410 Myr ago), the oldest spider silk record dates back to the Earliest Cretaceous (~140 Myr ago). The silk is preserved in a piece of amber, which was found within alluvial soils of the Ashdown Formation, Hastings (UK) (*Brasier, Cotton & Yenney, 2009*; *GWR, 2017*).

(e) *Oldest web with entrapped prey—~110 Myr ago.* The oldest web with entrapped prey is preserved in a cylindrical stalactitic mass of amber, dating back to Early Cretaceous (around 110 Myr ago). The fossil sample was discovered in San Just (Spain), and contains 26 strands of sticky silk entrapping a beetle, a parasitic wasp, a mite and a fly (*Peñalver, Grimaldi & Delclòs, 2006*; *GWR, 2017*).

(f) *Oldest recorded predatory event—~100 Myr ago. Poinar & Buckley (2012)* recently published the description of an Early Cretaceous Burmese amber of *~100 Myr ago*, containing a fossil spider (*Geratonephila burmanica* Poinar) in the process of attacking its ensnared prey, the parasitic wasp *Cascoscelio incassa* (Hymenoptera: Platygasteridae). While amber contains numerous examples of insects entrapped in spider webs (see, e.g., "*Oldest web with entrapped prey*"), there was no previous fossil record documenting such a predatory behavior (*GWR, 2017*).

(g) *Oldest social spider—Geratonephila burmanica Poinar (Araneidae).* The amber cited in the previous record ("*Oldest recorded predatory event*") contained both a male and juvenile spider sharing the same web. According to *Poinar & Buckley (2012)* this fossil record thus provides the first evidence of sociality in spiders. Since extant male

nephilids live in female webs, the presence of a male indirectly implies the presence of a female. Given that sociality in spiders involves the coexistence of adults and juveniles in a common web, it may be that the species was social.

(h) *Largest fossil spider*—Mongolarachne jurassica *(Selden, Shih and Ren) (Mongolarachnidae)*. With a body length of ~2.5 cm and first legs of nearly 6 cm, *Mongolarachnidae jurassica* from Middle Jurassic (~165 Myr ago) found in the strata of Daohugou in Inner Mongolia is the largest known fossil spider know to date (*Selden, Shih & Ren, 2011*, *2013*).

## Taxonomy and systematics

(a) *First spider(s) ever described in binomial nomenclature—Shared by 68 species.* The record for the first spider ever described in binomial nomenclature is shared by 68 species described by Carl Alexander Clerck in 1757. Actually, some of them are nowadays considered doubtful species, leaving the total to 53 currently valid species (*WSC, 2017*). These species own a second record, being among the first animals ever described using the binomial system of nomenclature (see also "*First arachnologist in history*") (Fig. 2A).

(b) *First listed spider alphabetically*—Abacoproeces molestus *Thaler (Linyphiidae)*. *Abacoproeces molestus* is the first valid spider species listed alphabetically in *WSC (2017)*. It is worth mentioning that *Abacoproeces brunneipes* (Dahl) would be the first spider name listed alphabetically, but this species is currently considered a junior synonym of *Abacoproeces saltuum* (L. Koch) (*WSC, 2017*).

(c) *Last listed spider alphabetically*—Zyuzicosa zeravshanica *Logunov (Lycosidae)*. *Zyuzicosa zeravshanica* is the last spider species listed alphabetically in *WSC (2017)*.

(d) *Longest scientific name*—Dipoena santaritadopassaquatrensis *Rodrigues (Theridiidae)*. This spider's name has 33 characters. Names with 32 characters are more common, such as *Alloclubionoides wolchulsanensis* (Linyphiidae), *Anophthalmoonops thoracotermitis* (Oonopidae), *Mecysmaucheniodides nordenskjoldi* (Mecysmaucheniidae), *Megalepthyphantes pseudocollinus* (Linyphiidae), and *Troglohyphantes typhlonetiformis* (Linyphiidae).

(e) *Shortest scientific name*—Gea eff *Levi (Araneidae)*. This spider has only six characters in its name. Names with seven characters are found in the genus *Ero* (Mimetidae) and *Copa* (Corinnidae).

(f) *Largest spider family—Jumping spiders, family Saltidicae.* The largest spider family is Salticidae with more than 6,000 species currently described, distributed in 634 genera (*WSC, 2017*; see also *GWR, 2017*).

(g) *Smallest spider family—Huttoniidae and Trogloraptoridae.* The smallest families of spiders are Huttoniidae and Trogloraptoridae, both of which include one single species—*Huttonia palpimanoides* O. Pickard-Cambridge and *Trogloraptor marchingtoni* Griswold, Audisio and Ledford, respectively. *Huttonia palpimanoides* is

endemic to New Zealand (*Paquin, Vink & Dupérré, 2010*) while *Trogloraptoridae marchingtoni* was discovered in few caves in southwestern Oregon, USA (*Griswold, Audisio & Ledford, 2012*).

(h) *First entire genome sequenced*—Stegodyphus mimosarum *Pavesi (Eresidae).* In 2014, the entire genome (the complete set of genetic material in an organism) of the African social velvet spider *Stegodyphus mimosarum* was sequenced for the first time by *Sanggaard et al. (2014)*. In the same study, the author published the first draft assembly of the genome of the mygalomorph spider *Acanthoscurria geniculata* (C.L. Koch) (Theraphosidae). The estimate genome size for *Stegodyphus mimosarum* is 2.55 gigabases (Gb; where 1 Gb is $10^9$ base pairs), whereas for *Acanthoscurria geniculata* is 6.5 Gb. Conversely, the first entire genomes of orb-weaving spiders [*Nephila clavipes* (Linnaeus) (Araneidae) and *Parasteatoda tepidariorum* (C.L. Koch) (Theridiidae)], were obtained in 2017. The estimated size for *N. clavipes* was 3.45 Gb (*Babb et al., 2017*) and for *Parasteatoda tepidariorum* 1.44 GB (*Schwager et al., 2017*). For comparison, the estimated human genome size is around 3 Gb (*International Human Genome Sequencing Consortium, 2004*).

(i) *Most species named after celebrities within one genus—Caribbean* Spintharus *species (Theridiidae).* The International Code of Zoological Nomenclature (*ICZN, 1999*) gives the taxonomist no specific rules on how to name new species. Thus, unsurprisingly, many taxa have been named after famous scientists and celebrities, or mythological, biblical and pop-cultural characters (*Jóźwiak, Rewicz & Pabis, 2015*). These homages to celebrities often attract a lot of attention from social media. As far as spiders are concerned, taxonomists have been inspired by well-known literature characters and writers [e.g., The Jungle Book by Rudyard Kipling: *Bagheera kiplingi* (Saticidae) (*Peckham & Peckham, 1896*); Harry Potter books by J.K. Rowling: the hat-looking spider *Eriovixia gryffindori* (Araneidae) (*Ahmed, Khalap & Sumukha, 2016*); Orson Welles and William Shakespeare: *Orsonwelles macbeth* (Linyphiidae) (*Hormiga, 2002*)], by actors, actress and movie characters [e.g., Angelina Jolie: the trapdoor spider *Aptostichus angelinajolieae* (Euctenizidae) (*Bond & Stockman, 2008*); Terminator: *Hortipes terminator* (Corinnidae) (*Bosselaers & Jocqué, 2000*); Predator and Arnold Schwarzenegger: *Predatoroonops schwarzeneggeri* (Oonopidae) (*Brescovit et al., 2012*)], and even by singers and progressive rock bands [e.g., Pink Floyd: long-jawed spiders in the genus *Pinkfloydia* (Tetragnathidae) (*Dimitrov & Hormiga, 2011*); Johnny Cash: *Aphonopelma johnnycashi* (Theraphosidae) (*Hamilton, Hendrixson & Bond, 2016*); David Bowie: *Heteropoda davidbowie* (Sparassidae) (*Jäger, 2008*); Neil Young: *Myrmekiaphila neilyoungi* (Euctenizidae) (*Bond & Platnick, 2007*)], among other examples. To date, the record for the spider genus with most species dedicated to celebrities goes to smiley-faced spiders *Spintharus* (Theridiidae). Recently, 15 new species from the Caribbean region were named after very famous people who stood up for human rights and were committed to nature conservation, including David Attenborough (*Spintharus davidattenboroughi*), Barack Obama and his wife (*S. barackobamai* and *S. michelleobamaae*), David Bowie (*S. davidbowiei*),

Leonardo Di Caprio (*S. leonardodicaprioi*) and Bernie Sanders (*S. berniesandersi*) (*Agnarsson et al., 2017*).

## Anatomy
### *Size*

(a) *Largest living spiders—Theraphosa blondi (Latreille) (Theraphosidae) and* Heteropoda maxima *Jäger (Sparassidae).* The Goliath bird-eater, *Theraphosa blondi* is possibly the largest known spider by mass (Fig. 3A). According to *GWR (2017)*, a single reared individual reached a leg span of 28 cm and a weight of 170 g. The giant huntsman spider, *Heteropoda maxima* (Sparassidae), discovered from caves in Laos, is possibly the largest known spider by leg span (up to 30 cm; *Jäger, 2001*; Fig. 3B). With a total body length up to 39.7 mm and a leg span of over 10 cm, females of *Nephila komaci* Kuntner and Coddington (Araneidae) are the largest known orb-web spiders (*Kuntner & Coddington, 2009*).

(b) *Smallest adult female spider—Anapistula ataecina Cardoso & Scharff (Symphytognathidae).* The record for the smallest adult female spider goes to one specimen of the type series of *Anapistula ataecina*, with a body length of 0.43 mm. The species was discovered in the Frade cave system (Portugal); the male is still unknown (*Cardoso & Scharff, 2009*).

(c) *Smallest adult male spider—Patu digua Forster and Platnick (Symphytognathidae).* With a total length of about 0.37 mm (not including the chelicerae), *Patu digua* is the smallest adult male spider ever described (*Forster & Platnick, 1977*). Instead, the *GWR (2017)* reports the congeneric *P. marplesi* Forster as the smallest spider (0.3 mm). However, according to the original description, the male of *P. marplesi* has a prosoma length of 0.22 mm, and an abdomen of 0.21 mm (*Forster, 1959*), in contrast to 0.15 mm and 0.25 mm (prosoma and abdomen, respectively) in *P. digua* (*Forster & Platnick, 1977*). Intra-specific variability in the body size is possibly at the base of this discrepancy. It is also worth noticing, that there are other spiders of similar size for which only the female is described (see, e.g., "*Smallest adult female spider*").

(d) *Most extreme sexual size dimorphism—Females weighing 125 times that of males.* Sexual size dimorphism is a morphological syndrome in which conspecific male and female sizes differ significantly. Among terrestrial animals, the most extreme female-biased gigantism is found in orb-weaving spiders (*Foellmer & Moya-Laraño, 2007*). Golden orb-weaving spiders (Araneidae) are the most extremely sexually size dimorphic. Female on average can be up to 125 heavier than mating males (*Kuntner et al., 2012*) (Fig. 3G).

(e) *Most unusual sexual size dimorphism—Males larger than females.* In most web-building spiders, females are larger than males (see "*Most extreme sexual size dimorphism*"). The water spider *Argyroneta aquatica* (Clerck) (Cybaeidae) (Fig. 4F) is one of the few spiders in which males are larger than females, possibly showing the most extreme male-biased sexual size dimorphism among spiders (*Schütz & Taborsky, 2003*, *2005*;

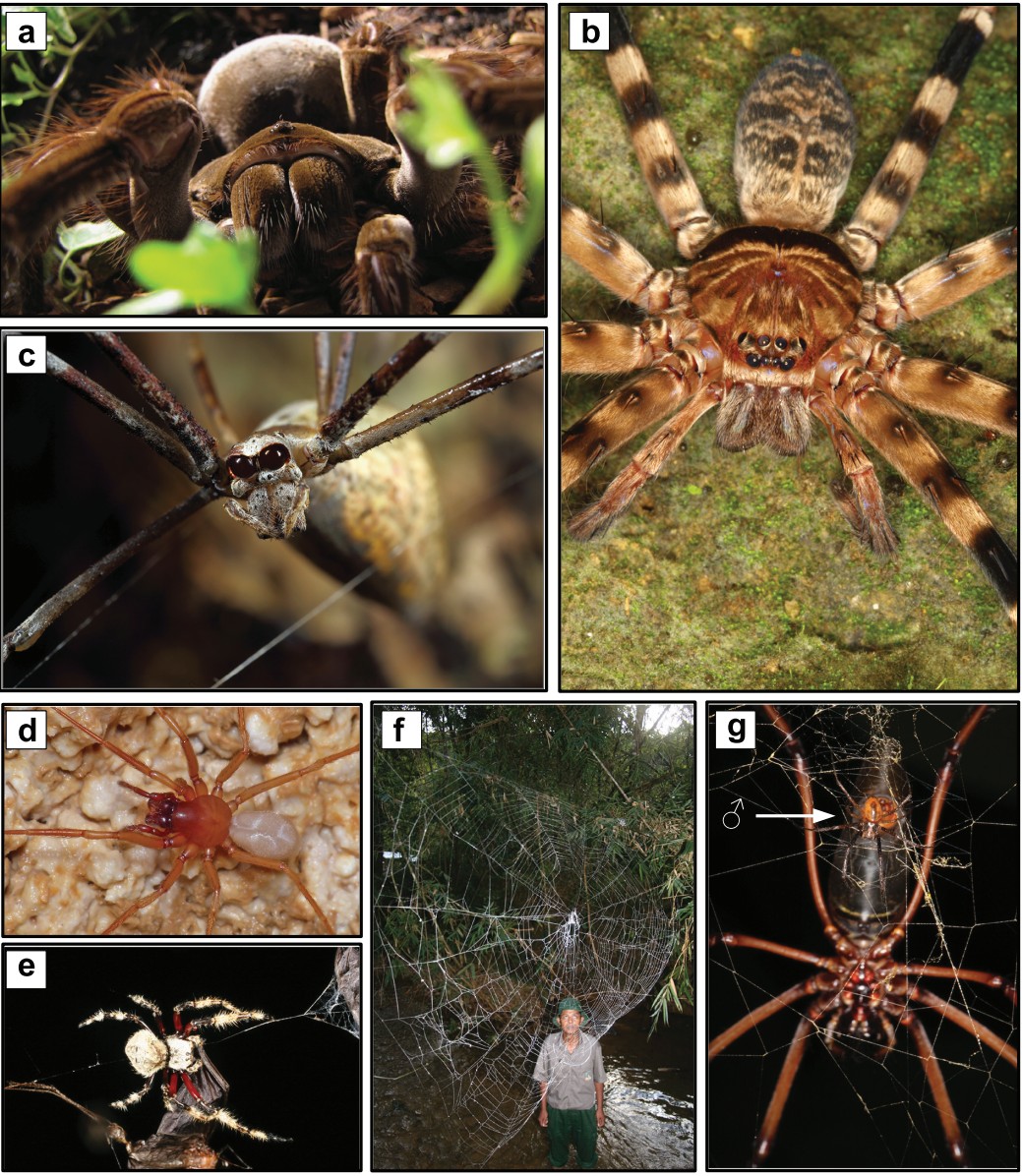

**Figure 3 Morphology and physiology.** (A) The Goliath bird-eater, *Theraphosa blondi* (Latreille) (Theraphosidae), the largest known spider by mass (Photo credit: Steve Le Roux). (B) *Heteropoda maxima* Jäger (Sparassidae), the largest known spider by leg span, in its typical ambushing position (Photo credit: Peter Jäger). (C) The enlarged posterior median eyes of a net-casting spider (*Deinopis* sp., Deinopidae) (Photo credit: Michael Doe). (D) *Stalita taenaria* Schiödte (Dysderidae), the first eyeless spider ever described (Photo credit: Fulvio Gasparo). (E) The Darwin's bark spider, *Caerostris darwini* Kuntner & Agnarsson (Araneidae), produces the toughest known spider silk (Photo credit: Matjaž Kuntner). (F) The web of the Darwin's bark spider can reach an area of 2.8 m$^2$, being therefore the largest orb-web ever measured (Photo credit: Matjaž Kuntner). (G) Golden orb-weaving spiders (Nephilidae) exemplify the most extreme male-biased sexual size dimorphism in spiders. The white arrow points at the male (Photo credit: Matjaž Kuntner).

*Seymour & Hetz, 2011*). It has been suggested that larger males should have mobility advantages over smaller ones when moving under water (*Schütz & Taborsky, 2003*). See also "*Most bioluminescent*" for another case of unusual sexual dimorphism.

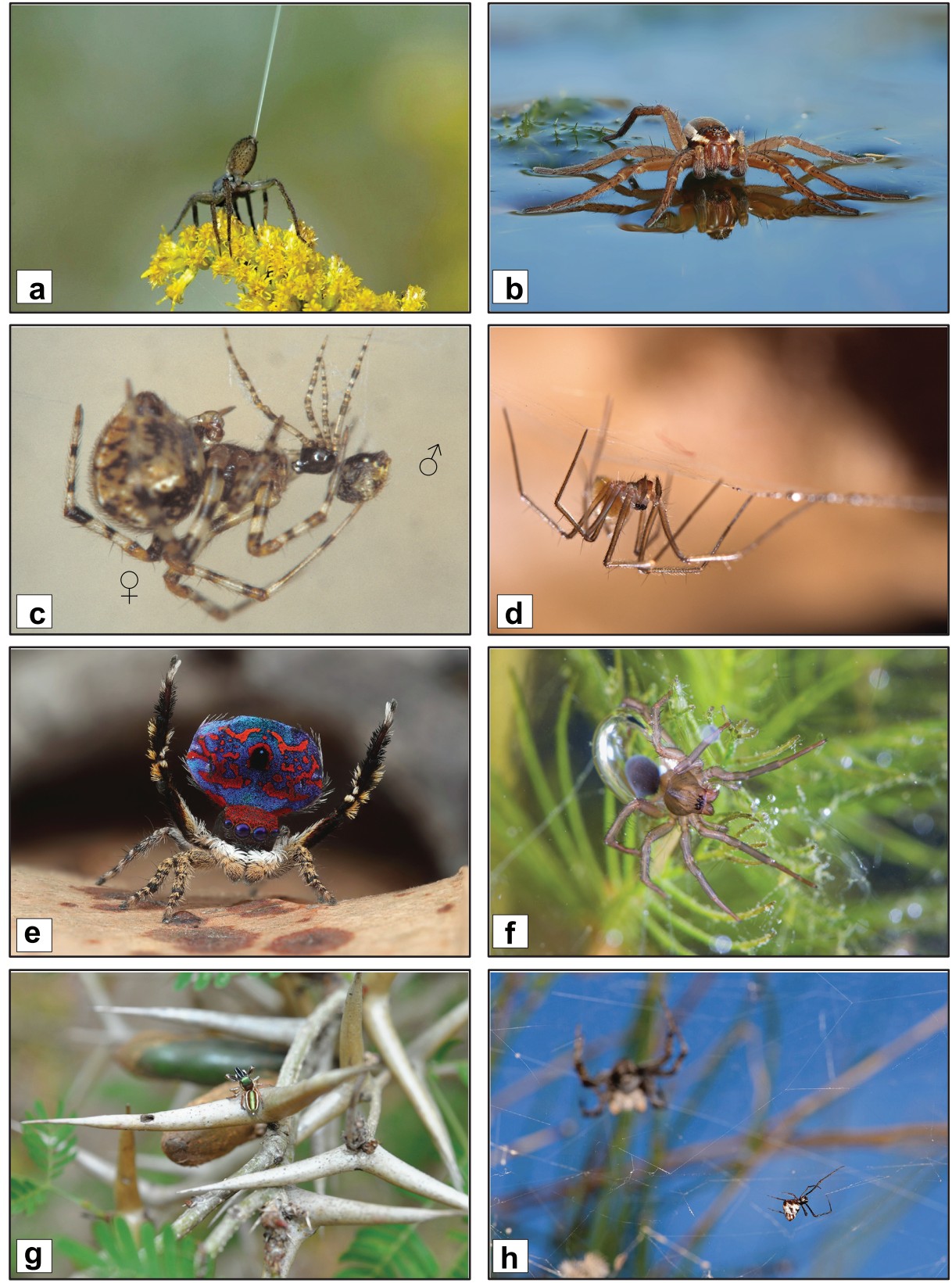

**Figure 4 Ecology and behavior.** (A) A ballooning spider—numerous spiders can disperse through the air by releasing one silk thread to catch the wind (Photo credit: Lacey Szymanski—Pieceoflace photography). (B) A fishing spider, *Dolomedes* sp. (Pisauridae), capable of effective locomotion on the surface of water (Photo credit: Olaf Craasmann). (C) A male and female of the one-palped spider *Tidarren argo* Knoflach & van Harten (Theridiidae) during the copula: in this species, the male dies almost immediately after the insertion of his copulatory organ and is usually cannibalized by the female afterwards (Photo credit: Barbara Knoflach-Thaler). (D) A cave-dwelling spider of the genus *Troglohyphantes*. In some species, a protracted mating lasting >18 hours was observed (Photo credit: Francesco Tomasinelli). (E) A male of *Maratus madelinae* Waldock (Salticidae) performing its courtship display (Photo credit: Michael Doe). (F) The water spider, *Argyroneta aquatica* (Clerck) (Cybaeidae), the only known spiders living a wholly aquatic life (Photo credit: Riccardo Cavalcante). (G) *Bagheera kiplingi* Peckham & Peckham (Salticidae), the only known spider with a mostly herbivorous diet—it predominantly consumes specialized leaf tips of *Acacia* (Photo credit: Maximilian Paradiz via Wikipedia). (H) A kleptoparasitic spider (Theridiidae: *Argyrodes* sp.) dwelling in the web of a Tropical Tent-Web Spider, *Cyrtophora citricola* (Araneidae) (Photo credit: Emanuele Biggi).

## Body parts

(a) *Highest number of eyes—Eight*. The highest number of eyes in spiders is eight, as found in countless species. An anecdotic record is held by *Troglohyphantes polyophthalmus* Joseph (Linyphiidae), which possesses sixteen eyes according to the original description—as emphasized by the specific epithet (*Joseph, 1881*). However, this species was described on a specimen killed in the early stage of molting, so that the number of eyes appeared doubled.

(b) *Least number of eyes—Zero*. The first eyeless spider ever described is *Stalita taenaria* Schiödte (Dysderidae) from the Postojnska cave in Slovenia (*Schiödte, 1847*). *Stalita taenaria* shares the record for the least number of eyes (zero) with more than 1,000 eyeless spider species inhabiting caves and other subterranean habitats around the world (*Mammola & Isaia, 2017*) (Fig. 3D).

(c) *Largest eyes—Net-casting spiders, family Deinopidae*. Net-casting spiders in the genus *Deinopis* possess extraordinary enlarged posterior median eyes (*Blest & Land, 1977*), possibly the largest eyes of a spider (up to 1.4 mm in diameter; *GWR, 2017*). These eyes all possess large photoreceptors (20 μm wide, 110 μm long; *Land & Nilsson, 2012*), which are crucial for gathering light for nocturnal vision (see "*Best nocturnal eyesight*") (Fig. 3C).

(d) *Longest relative chelicerae—Assassin spiders, family Archaeidae*. Different spider species across many families exhibit chelicerae elongation, such as long-jawed spiders (Tetragnathidae), long-jawed intertidal spiders (Desidae) and some jumping spiders (Salticidae). However, the highest ratio chelicerae/body size is possibly found in the assassin spiders—also known as pelican spiders. In many species, the length of the chelicerae almost equal the body length. Assassin spiders are cursorial hunters, specialized to feed upon other spiders. They are unique in their extreme modification of the cephalic area and jaws, giving them the appearance of a "neck" and "head" (*Rix & Harvey, 2011*; *Wood, Griswold & Gillespie, 2012*).

(e) *Largest relative fangs—males of* Myrmarachne, *family Salticidae*. In relation to their body size, the males of ant-mimicking spiders in the genus *Myrmarachne* (family Salticidae) not only have large chelicerae, but also extremely long fangs exceeding the length of the prosoma [see illustration in *Millot (1949a*, p. 602, f. 369*)*]. The hypertrophy of the fangs is a male secondary sexual character, whereas females have normal fangs (*Millot, 1949a*).

(f) *Longest relative walking legs—Unclear.* Being the largest living spider by leg span, the giant huntsman spider *Heteropoda maxima* Jäger (Sparassidae) is most likely the species with the absolute longest walking legs (see "*Largest living spiders*"; Fig. 3B). However, if the length of the legs is relativized to the body length, probably the longest legs are found in the daddy long-legs spiders or house spiders (family Pholcidae) and in certain species of Ochyroceratidae (gen. *Althepus* and *Leclercera*). In species belonging to these groups, the length of the legs may be more than five to seven times the body size (see *Jocqué & Dippenaar-Schoeman, 2006*).

(g) *Most legs—10.* In insects, the expression domain of the Hox gene Antennapedia (Antp) controls the expression of legs. *Khadjeh et al. (2012)* used RNA inference to downregulate this gene in the spider *Achaearanea tepidariorum* (C. L. Koch) (Araneae), giving rise to a 10-legged phenotype, which is, therefore, the spider with the highest number of legs.

(h) *Most spinnerets—four pairs in* Mesothelae. The spiders with the greatest number of spinnerets are those belonging to the suborder Mesothelae (*GWR, 2017*), which includes one living family (Liphistiidae) and a number of fossil representatives (*Dunlop, Penney & Jekel, 2017*; *WSC, 2017*). They possess four pairs of spinnerets, positioned in the middle of their segmented abdomen (*Haupt, 2003*). All other spiders possess from one to three pairs of spinnerets.

(i) *Longest relative spinnerets—Long-spinnered spiders, family Hersiliidae.* Extremely elongated posterior spinnerets can be found in representative of the family Hersiliidae (see *Jocqué & Dippenaar-Schoeman, 2006*). In certain species, they can be as long as the body of the spider (prosoma + opisthosoma). The enlarged anterior lateral spinnerets of Molycriinae (a subfamily of the long-spinneret ground spiders Prodidomidae) are also remarkable being tube-like and extending throughout the whole opisthosoma (*Platnick & Baehr, 2006*).

### Internal organs

(a) *Largest central nervous system—Very small spiders.* The internal anatomy has been studied in details in very few species, and thus it is difficult to assess which species has the largest—or the smallest—central nervous system (CNS). Recent allometric studies of the gross neuroanatomy of a number of spider species, shown that very small spiders (including nymphal stages) have disproportionately larger CNSs relative to body mass when compared with large-bodied forms. In fact, the brains of small spiders may extend out of their body cavity into their walking legs (coxae) (see *Quesada et al., 2011*, p. 526, f. 4). Accordingly, the relatively large CNS of very small spiders can occupy up to 78% of the cephalothorax volume (*Quesada et al., 2011*).

(b) *Largest relative venom glands—Filistatidae and Plectreuridae.* Based on the comparative studies by *Millot (1949b)*, two main venom gland organisations can be distinguished—cylindrical and multilobular glands. Cylindrical glands can be short as in Mesothelae and most orthognath spiders, but can extend far into the prosoma as in

most araneomorph spiders. According to the *GWR (2017)*, the absolute largest venom glands are those of the wandering spiders of the South American genus *Phoneutria* (Ctenidae), measuring up to 10.4 × 2.7 mm (see also "*Most venomous to humans*"). However, when considering the largest surface area relativized to body size, the multilobular glands reported for Filistatidae [*Filistata insidiatrix* (Forsskål)] and Plectreuridae (*Plectreurys* sp.) seems to be the most conspicuous (*Millot, 1949b*). For example, the large multilobular venom glands of *F. insidiatrix* occupy half of the prosoma as depicted by *Millot (1949b*, f. 438*)*.

(c) *Smallest relative venom glands—Mesothelae.* The smallest venom gland (relative size) is reported from the most basal branching spider group, the Mesothelae. The gland of these ancient spiders extends only slightly behind the articulation of the fang and is very small and inconspicuous (*Bristowe & Mollot, 1933*; *Foelix & Erb, 2010*). This might be also the reason why Mesothelae were thought to lack venom glands (*Haupt, 2003*). In fact, venom glands are part of the ground pattern of spiders and only in the family Uloboridae they are absent (see "*Least venomous spiders*").

## Physiology

### Silk and webs

(a) *Largest web (area)—2.8 m².* The Darwin's bark spider, *Caerostris darwini* Kuntner and Agnarsson (Araneidae) spins a web whose surface ranges from 0.09 to 2.8 m². The largest measured web in this species was about 2.8 m², being therefore the largest orb-web ever measured (*Kuntner & Agnarsson, 2010*; *GWR, 2017*). Prior to the discovery of this species, the record was held by representatives of the genus *Nephila* (Araneidae), capable of spinning orb-webs of more than 1 m diameter (*Kuntner & Coddington, 2009*) (Figs. 3E and 3F)

(b) *Largest web (length)—25 m.* The anchor lines of the web of Darwin's bark spider, *Caerostris darwini* Kuntner & Agnarsson (Araneidae), are capable of bridging over 25 m, being the longest web among all spiders (*Kuntner & Agnarsson, 2010*; *Gregorič et al., 2011*; *GWR, 2017*).

(c) *Smallest web—less than 10 mm.* The smallest spider webs are spun by representatives of the family Symphytognathidae (see "*Smallest adult male spider*" and "*Smallest adult female spider*"). According to estimations, in many species the webs can be less that 10 mm in diameter (*GWR, 2017*).

(d) *Strongest silk—520 MJ/m³.* The Darwin's bark spider, *Caerostris darwini* Kuntner & Agnarsson (Araneidae), produces the toughest known spider silk (*GWR, 2017*; see also "*Largest web*"). Tensile testing has shown that certain threads may reach the toughness of 520 MJ/m³ (average = 350 MJ/m³). The silk of *Caerostris darwini* is therefore over 10 times tougher than Kevlar® (*Agnarsson, Kuntner & Blackledge, 2010*; *GWR, 2017*) (Fig. 3E).

(e) *Strongest cocoon silk—Maximum stress = 0.64 GPa and strain = 751%.* The record for the most stretchable egg sac silk goes to the stalk silk of the cocoon of *Meta menardi* (Latreille) (Tetragnathidae), for which tensile testing pointed out a maximum stress and strain of 0.64 GPa and 751%, respectively (*Lepore et al., 2012*). On the other hand,
the toughness of the egg case silk threads recorded to date ($G = 193$ MJ m$^{-3}$) is spun by the hermit spider *Nephilengys cruentata* (Fabricius) (Araneidae) (*Alam et al., 2016*).

## Venom

(a) *Most venomous to humans—Australian funnel-web spiders, family Hexathelidae.* In general, only few spider taxa are renowned for the efficacy of the venom, e.g., widow spiders (*Latrodectus* spp.; Theridiidae) causing latrodectism and recluse spiders (*Loxosceles* spp.; Sicariidae) causing severe skin lesions and systemic effects. Wandering spiders of the South American genus *Phoneutria* (Ctenidae) are known to be very poisonous by transferring large quantities of a strong neurotoxin during a single bite. However, it is important to emphasize that verified bites from other spider species cause only minor and transient effects (*Vetter & Isbister, 2008*). Australian funnel-web spiders (family Hexathelidae) are considered to be the most dangerous spiders (to humans) in the world (*White, 2000*; *Isbister et al., 2005*). Within the family, the most venomous spider is possibly the Sydney funnel-web spider male, *Atrax robustus* O. Pickard-Cambridge. In this species, just 0.2 mg/kg of the venom of the male is lethal for humans (*GWR, 2017*). On the other hand, according to literature reviews (*Isbister et al., 2005*), the tree-dwelling Australian funnel web spider *Hadronyche cerberea* L. Koch has the highest rate of severe envenomations (75%), in contrast to 17% in *A. robustus*. Since the development of antidotes against funnel-web spider envenomation, no fatal bites have been reported (*Nentwig & Kuhn-Nentwig, 2013b*).

(b) *Least venomous—Shared by two families.* Least venomous spiders are representative of the families Holarchaeidae and Uloboridae. Holarchaeidae entirely lacks openings of the poison glands (*Kuhn-Nentwig, Stöcklin & Nentwig, 2011*; *Nentwig & Kuhn-Nentwig, 2013a*), whereas Uloboridae entirely lack cheliceral venom glands (*GWR, 2017*). The latter have evolved an alternative hunting strategy: they wrap their prey in silk, cover it in regurgitated digestive enzymes and toxins and then ingest the liquified body (*Weng, Barrantes & Eberhard, 2006*).

(c) *Most unusual venom—Spitting spiders, family Scytodidae.* Spitting spiders produce the most unusual spider venom type. They have a domed cephalothorax that houses a large pair of glands, producing a mixture of venom and glue (*Foelix, 2011*). This mixture plays a crucial role in their unique prey capturing technique (see "*Most creative hunting strategies*"). The components expressed in the venom glands of one of the most common species of spitting spiders [*Scytodes thoracica* (Latreille)] have been recently identified by *Zobel-Thropp et al. (2013)*. These include homologues of toxic proteins astacin metalloproteases, venom allergen, longistatin, and translationally controlled tumor protein.

## Sensory organs

(a) *Best diurnal eyesight—Jumping spiders, family Salticidae.* Despite the majority of spiders possess eight eyes, most species are known to have poor eyesight (*Foelix, 2011*).

This is especially true in web-spinning spiders, relying mostly on vibrational cues for foraging and mating, rather than on visual perception. A notable exception is found in jumping spiders (Salticidae), being diurnal active ground-dwellers renowned for their high performing visual system (*Jackson & Pollard, 1996*; *Zurek & Nelson, 2012*; *Menda et al., 2014*). They possess enlarged principal eyes which are specialized for resolution vision, functioning like moveable telescopes (*Land, 1969*). In addition, the three pairs of secondary eyes are highly sensitive to motion and collectively encompass a 360° field of view (*Duelli, 1978*; *Zurek et al., 2010*; *Zurek & Nelson, 2012*). Jumping spiders use this pronounced visual acuity in hunting and mating (see, e.g., "*Most specialized prey classification*," "*Best hearing*" and "*Most elaborate courtship*").

(b) *Best nocturnal eyesight—Net-casting spiders,* Deinopis *spp. (Deinopidae).* The best nocturnal eyesight documented to date is found in the net-casting spiders (*Deinopis* spp.). They possess enlarged posterior median eyes (see also "*Largest eyes*") that are reported to be 2,000 times more sensitive to light than human eyes, thus appearing physiologically designed for detecting movement at night (*Blest & Land, 1977*). It has been suggested that these visual cues are fundamental to net-casting spiders for capturing cursorial prey items (*Stafstrom & Hebets, 2016*; see "*Most creative hunting strategies*") (Fig. 3C).

(c) *Best hearing—Jumping spiders, family Salticidae.* *Shamble et al. (2016)* recently presented behavioral and neurophysiological evidences about airborne sounds perception by jumping spiders. They reported that jumping spiders are able to perceive and respond to airborne acoustic stimuli, even at relatively large distances of about 3 m. Behavioral experiments revealed that the jumping spider *Phidippus audax* Hertz is able to respond even to low-frequency sounds (around 80 Hz). However, very few spider species have been tested in this respect.

(d) *Most bioluminescent—Cosmophasis umbratica Simon (Salticidae).* The jumping spider *Cosmophasis umbratica*, distributed from India to Indonesia (*WSC, 2017*), is the only known spider for which ultraviolet (UV) reflectance and the ability to see UV have been demonstrated experimentally (*Lim & Li, 2006a*, *2006b*; *GWR, 2017*). This species is sexually dimorphic in the reflectance of UV, with males having UV-reflecting markings and females displaying UV-induced green fluorescence. The bioluminescence in this species is crucial for the success of mating (*Lim, Land & Li, 2007*).

### Biological cycle

(a) *Longest life span—~40 years.* In spiders, data about life span in the wild are extremely scarce. It was assumed that the enigmatic Tasmanian cave spider, *Hickmania troglodytes* (Higgins & Petterd) (Austrochilidae), reaches a life span of several decades (*Doran et al., 1999*). The greatest longevity documented is found in Theraphosidae in captivity, with certain species having a life expectancy of more than 30 years (data on *Theraphosa* and *Aphonopelma*; *Schultz & Schultz, 1998*; *Ibler, Michalik & Fischer, 2013*; *GWR, 2017*).

(b) *Shortest circadian rhythm—19 h.* *Moore et al. (2016)* recently described behavioral rhythms of locomotor activity and web building in the orb-weaving spider *Cyclosa*

*turbinata* (Walckenaer) (Araneidae). They discovered that this species yield an exceptionally short-period clock, diverging from the natural 24-h light/dark cycle. In this species, the period of the free run is about 19 h.

### Eggs and sperm

(a) *Longest sperm—0.65 mm.* The longest known spider sperm by far is reported for the goblin spider *Neoxyphinus termitophilus* (Bristowe) (Oonopidae). With approximately 0.65 mm, one sperm measures around 1/3 of the body length of this spider (*Lipke & Michalik, 2015*). The sperm is transferred coiled and encapsulated in groups resembling the so-called synspermia, which have a diameter of approximately 0.07 mm in this species. The longest transfer form (0.08 mm) is held by another goblin spider, *Orchestina* spp. (*Lipke & Michalik, 2015*).

(b) *Highest number of eggs—>3,000.* The number of eggs laid by spiders is highly variable depending on species and female body mass (*Marshall & Gittleman, 1994*). *Robinson & Robinson (1976)* reported more than 2,400 eggs for a species of *Nephila* (Araneidae). The same authors estimated that for other *Nephila* species one female may produce as many as 3,000 eggs in multiple egg sacs. According to available evidences, *Nephila pilipes* (Fabricius) is possibly the spider capable to lay the highest number of eggs per clutch. In this species, the egg sac usually contains more than 3,000 eggs (*Higgins, 2002*). Higgins observed how female fecundity (number of eggs laid per clutch) is a function of pre-laying female mass (see *Higgins, 2002*, p. 382). The mass of the largest female sampled by Higgins was 6.9 g and so, it is possible to estimate that the clutch size of this female should be equivalent to 9,724 eggs. On the other hand, the *GWR (2017)* reports up to 3,000 eggs for a species apparently belonging to the genus *Mygalomorphus*, which is also deemed to lay the largest eggs (having the size of a small pea). However, despite being reported in a few websites, *Mygalomorphus* is neither a valid name nor a synonym in spider fossil (*Dunlop, Penney & Jekel, 2017*) or extant (*WSC, 2017*) nomenclature.

(c) *Least number of eggs—One.* The *GWR (2017)* currently reports *Oonops domesticus* Dalmas (Oonopidae) as the spider laying the fewest number of eggs, namely two eggs for each clutch. However, in *Telema tenella* (Simon) (Telemidae), a European cave-dwelling spider, the lowest number of eggs found in a single eggsac is one (*Juberthie, 1985*). The tendency to lay small numbers of eggs is a well-known adaptation to subterranean habitats. Studies on subterranean spiders are however scarce: it is likely that *Telema tenella* may share this record with other cave species for which the number of eggs/cocoon was never quantified (*Mammola & Isaia, 2017*).

## Behavior
### Locomotion

(a) *Best ballooners—Most spiders.* Many spiders, especially small species or immature stages, disperse by releasing one or more silk threads to catch the wind (the so-called ballooning

behavior). Airborne dispersal is particularly widespread amongst higher Entelegyne spiders (*Bell et al., 2005*). Distances travelled by spider ballooners can reach >1,000 km, as testified by sailors who reported spiders caught in their ships in the middle of oceans (*Bell et al., 2005*). Possibly, the longest distance covered with ballooning is 3,200 km, as reported by *Gressitt (1965)* for an unidentified linyphiid spider (Fig. 4A).

(b) *Best sailors—Fishing spiders (Pisauridae).* The ability to walk on water has evolved independently among over 1,200 species of vertebrates and invertebrates (*Bush & Hu, 2006*). Spiders in many families are capable of locomotion on the surface of water (*Suter, 2013*). Most likely, the best sailors are the adults of fishing spiders *Dolomedes* spp. (Pisauridae), capable of moving across water surfaces taking advantage of wind currents (*Suter, 1999*). More recently, it was demonstrated that ballooning linyphiids and tetragnathids also display sailing-related behaviors, as specific responses to landing on water surfaces after ballooning (*Hayashi et al., 2015*; see also "*Best ballooners*") (Fig. 4B).

(c) *Fastest spider—Cebrennus rechenbergi Jäger (Sparassidae).* The *GWR (2017)* reports the giant house spider *Eratigena atrica* (Koch) (Agelenidae) (formerly known as *Tegenaria gigantea*) as the fastest spiders, having a maximum running speeds of 0.52 m/s (1.9 km/h). However, the flic-flac behavior of the Maroccan flic-flac spider *Cebrennus rechenbergi* is possibly the fastest locomotory behavior documented for spiders [for a description see *Jäger (2014*, p. 350, f. 152–161*)*]. It was interpreted as a last resort escaping behavior, by which the speed of the spider can increase up to two times the normal running speed (2 m/s according to estimations). This striking locomotory behavior has also inspired the construction of a robot with similar motional elements (*King, 2013*).

(d) *Fastest rotational movement—Flattie spiders, genus* Selenops. Flattie spiders are unique in their ability of performing rapid strike maneuvers to capture prey approaching from an unlimited range of directions. This extraordinary ability, documented by *Crews (2016)* at the last International Congress of Arachnology, is crucial for the success of their ambush striking. By reaching an angular speed of up to 3000 degrees per second, and completing all strikes in less than 120 milliseconds, they exhibit the fastest rotational movement in animals.

### Foraging

(a) *Most creative hunting strategies—Shared by different species.* Spiders are extremely creative in terms of hunting strategies. In the course of their evolution, many spider species have developed impressive hunting strategies, and thus the decision of which one is the most effective is subjective. Some of the most unusual are:

– Bolas spiders (Araneidae, Mastophorini) have evolved a hunting strategy that combines chemical mimicry (mimics pheromone blends to attract the prey) with a bolas-like weapon, which consist of a silk thread ending with a droplet of adhesive glue that the spider swing to catch its flying prey (*Yeargan, 1994*).

– Spitting spiders (*Scytodes* spp.), as their name suggest, have evolved a very peculiar hunting strategy to subdue prey: they spit a zig-zagged silken mixture of glue and venom to tether prey at a distance (see also "*Most unusual venom*"). Ejection velocities were measured as high as 28.8 m/s (*Suter & Stratton, 2009*).

– Net-Casting or Ogre Faced Spiders (*Deinopis* spp., *Deinopidae*) use their webs in a very unusual way. At night, net-casting spiders hang upside down, holding a rectangular capture silken snare, which is spun between their front legs. From this position, foraging spiders lunge toward prey, expanding the snare and actively entrapping aerial or terrestrial prey (*Robinson & Robinson, 1971*; *Stafstrom & Hebets, 2016*; see also "*Best nocturnal eyesight*" and Fig. 3C).

– The orthognath purse-web spiders (*Atypus* spp.; Atypidae) creates a tube of silk that is hidden partially underground, with the portion above ground being covered in leaves and other debris. The spiders waits upside-down in the aerial part of their silk tubes and impale prey (mainly insects) crawling over the tube with their large thin fangs. Afterwards, the impaled prey is dragged into the tube and once eaten the remaining parts of the prey are ejected through the opening at the top of the tube (*Enock, 1885*; *Bristowe, 1933*).

(b) *Fastest predatory strike—*Zearchaea sp. *(Mecysmaucheniidae)*. The fastest predatory strike in spiders was documented for trap-jaw spiders (family Mecysmaucheniidae). This family currently comprises 25 described species of tiny ground-dwelling spiders distributed in New Zealand and southern South America. Trap-jaw spiders rely on active hunting to prey capture. By means of high-speed video calculations, *Wood et al. (2016)* documented the speed of the power-amplified predatory strike in 14 species belonging to this family. The fastest was a species in the genus *Zearchaea*, capable of striking with a speed of 0.00012 s and releasing a power output of 60,000 W/kg (mean values of 3 recording events).

(c) *Largest invertebrate prey—Earthworms.* The largest invertebrate prey reported for spiders are giant earthworms up to 1 m in length (*Nyffeler et al., 2017*). These were consumed by *Theraphosa blondi* (see "*Largest living spiders*")

(d) *Largest vertebrate prey—Fish, toads, birds, bats.* Websites are full of stories and videos about spiders foraging on any kind of vertebrate animals. Although we acknowledge that some of them are truly impressive, we remain skeptical and rely on scientific literature. Accordingly, we report four scientifically documented cases of largest vertebrate prey:

– The largest fish captured by a spider is a goldfish *Carassius auratus* (Cyprinidae) of ~9 cm length and presumably 15 g weight. It was captured by a pisaurid spider in a garden pond in Sydney. However, under the assumption that the largest wandering spider, the ctenid *Ancylometes rufus* (Walckenaer) weighing up to 7 g, is as effective in overpowering oversized prey as the smaller-sized pisaurids, fish of up to 30 g might conceivably be killed in the wild (*Nyffeler & Pusey, 2014*).

– The largest amphibians captured by spiders are possibly toads. *Menin, de Jesus Rodrigues & de Azevedo (2005)* reported the predation of an individual of *Theraphosa blondi* (84.12 mm) (see "*Largest living spiders*") on a juvenile *Bufo marinus* (Bufonidae) of 90.52 mm length (see also "*Largest invertebrate prey*").

– According to *Brooks (2012)*, the largest bird found wrapped in a spider orb web is a laughing dove *Streptopelia senegalensis* (Columbidae) of 80 g (wing chord of 138 mm).

– The largest bat found wrapped in a spider web is a Gould's wattled bat, *Chalinolobus gouldii* (Vespertilionidae), weighing around 15 g (estimated value). It was captured by an unidentified web-building spider (*Nyffeler & Knörnschild, 2013*).

(e) *Strangest diet—Leaf tips.* Spiders are renowned to be carnivorous. Being the only spider (mostly) herbivorous, *Bagheera kiplingi* Peckham and Peckham (Saticidae), distributed from Mexico to Costa Rica, owns the record for the strangest diet (Fig. 4G). From behavioral observations and stable-isotope analysis, *Meehan et al. (2009)* showed that the diet of this spider predominantly comprises specialized leaf tips, the so-called Beltian food bodies. There are other spider species occasionally feeding on plant products (e.g., pollen), with at least 95 reports documented in literature (*Nyffeler, Olson & Symondson, 2016*) (Fig. 4G).

(f) *Fussiest spider—Evarcha culicivora* Wesolowska and Jackson (Salticidae). Prey specialization is uncommon in spiders (but see "*Longest chelicerae*" and "*Strangest diet*" for some examples). The jumping spider *Evarcha culicivora*, reported from western Kenia (*Wesolowska & Jackson, 2003*), is unique because it feeds indirectly on vertebrate blood by choosing blood-fed female mosquitoes (*Anopheles*) as prey (*Jackson, Nelson & Sune, 2005*). Studies have shown that *Evarcha culicivora* is able to discern between blood-fed mosquitoes from similar-sized prey that are not carrying blood, including congeneric male mosquitoes and females that have not fed (*Jackson, Nelson & Sune, 2005*; *Nelson & Jackson, 2006*, *2012*). Stemming from these observations, it has been suggested that this peculiar species may be useful for the biological control of malaria vectors (*Nelson & Jackson, 2006*).

### Reproduction

(a) *Shortest Mating—<1 s.* Given the wealth of literature and observations, it is not easy to decide about the shortest mating. If we consider mating as an interaction between two partners, the shortest ones are possibly found in a number of wasp spiders (*Argiope* spp.), in the one-palped spider *Tidarren argo* Knoflach and van Harten (Theridiidae) and in the dark fishing spider *Dolomedes tenebrosus* (Hentz) (Pisauridae). In fact, in these species, the male dies almost immediately after the insertion of his copulatory organ (spontaneous death) and is usually cannibalized by the female afterwards (*Foellmer & Fairbairn, 2003*; *Knoflach & van Harten, 2001*; *Schwartz, Wagner & Hebets, 2013*) (Fig. 4C).

(b) *Longest mating—>18 h.* In certain species of *Troglohyphantes* spider (cave-dwelling Linyphiids), *Deeleman-Reinhold (1978)* observed a protracted mating lasting >18 h.

However, due to the paucity of information, it seems likely that longer mating durations could be expected in other species (Fig. 4D).

(c) *Best date—Nuptial gifts in* Pisaura mirabilis *(Clerck) (Pisauridae).* "Nuptial gifts" are nutrients that males of a number of species (especially Arthropods) offer to females prior to, during, or shortly after copulation (*Gwynne, 2008*). In spiders, nuptial gifts have been documented in various forms as, e.g., glandular secretion or wrapped prey items. Possibly, the most spectacular nuptial gift is reported for *Pisauridae mirabilis* (e.g., *Van Hasselt, 1884*; but see *Itakura (1993)* for a possible other species), as it consists of large prey items wrapped up in silk by the male (*Prokop & Maxwell, 2012*)—but males may sometime 'cheat' by offering worthless gifts in term of nutrient content, e.g., by inflating their gifts with inedible items or excessive silk (*Ghislandi et al., 2017*). The male offers his nuptial gift during courtship and, while the female is feeding on it, he successfully mates with her. It has been suggested that the female's hunger state is thus decisive for mating success, as hungry females are more likely to accept a nuptial gift and hence to copulate (*Bilde et al., 2007*).

(d) *Most elaborate courtship—Jumping spiders, family Salticidae.* With a certain degree of subjectivity, the mating dance of peacock spiders *Maratus* spp. (Salticidae), can be listed among the most elaborate and beautiful courtship displays in arthropods (*Girard, Kasumovic & Elias, 2011*). Such "spider dance" recently received great attention in the social media—the videos about the courtship displays of different species of *Maratus* in the *Peacockspiderman* YouTube channel had cumulatively more than 12 million views as of June 2, 2017 (Fig. 4E).

(e) *Most complex song—Jumping spiders in the Habronattus coeacatus group.* During courtship, these spiders use complex multimodal signals made up of combinations of motion displays and vibratory songs. The latter are extremely complex, as they consist of up to 20 elements organized in functional groupings (motifs) that change as courtship progresses (*Elias et al., 2012*), thus possibly representing the most complex songs documented in spiders.

(f) *Loudest spider—Maratus michaelseni (Simon) (Salticidae).* Sound production by spiders has been documented in at least 26 families. Sounds are produced either by stridulation (friction of two body parts), or percussion (striking of the substratum). Sounds are used in at least three behavioral contexts: courtship, defense against predators and aggressive interactions between males (*Uetz & Stratton, 1982*). In certain species, sounds produced by spiders are even audible to the human ear, such as the one produced by *Anyphaena accentuata* (Walckenaer) (Anyphaenidae) and *Gladicosa gulosa* (Walckenaer) (Lycosidae) (*GWR, 2017*). To our knowledge, the loudest spider sound is produced by the males of the jumping spider *Maratus michaelseni* (Simon) (Salticidae). During courtship, this species produce sounds by stridulation, which can be heard several meters away (*Gwynne & Dadour, 1985*; see also "*Most elaborate courtship*").

(g) *Best mother—Matriphagy.* Providing offspring with food is thought to be the most important form of parental care. Possibly, the most "*unusual and extreme form of care*"

(*Evans, Wallis & Elgar, 1995*) is called matriphagy, in which the mother sacrifices herself to feed her offspring. This peculiar form of parental care has evolved at least in six spider families (*Schneider, 1996*).

(h) *Best father*—Dolomedes tenebrosus *(Hentz, 1844) (Pisauridae)*. In numerous spiders, females eat their mating partner just after the copula (see "*Shortest Mating*"). Such self-sacrifice is evolutionary advantageous if being eaten sufficiently increases offspring number or fitness (paternal effort hypothesis) or, either, the fertilization success. Recent experiments conducted by *Schwartz, Wagner & Hebets (2016)* on the dark fishing spider, *Dolomedes tenebrosus*, demonstrated an impact of male consumption on offspring size and overall survival indicating that self-sacrifice behavior should be adaptive.

### Lifestyle

(a) *Most peaceful*—Social spiders. The vast majority of spiders conduct a solitary lifestyle, and generally display an aggressive behavior even toward conspecifics. However, a small number of species have evolved different forms of group living lifestyles (*Lubin & Bilde, 2007*; see also "*Largest colony*"). Two main forms of sociality has arisen in spiders: (i) cooperative species ("social" *sensu Lubin & Bilde, 2007*) live in family group territories wherein they share communal nests and capture webs, which they inhabit together, cooperating in foraging and raising young; (ii) colonial species ("territorial permanent social" *sensu Avilés, 1997*) occur in aggregations, but individuals in the colony generally forage and feed alone and there is no maternal care beyond the egg stage. Among these two group living styles, the first is rare, being found in at least six families: Agelenidae, Dictynidae, Eresidae, Oxyopidae, Theridiidae, Thomisidae (*Lubin & Bilde, 2007*). On the other hand, coloniality is more common, being reported in at least 12 families (*Whitehouse & Lubin, 2005*). However, if considering species names, the record holders would be either *Singafrotypa mandela* Kuntner & Hormiga (Araneidae) and *Stasimopus mandelai* Hendrixson & Bond (Ctenizidae) or *Bristowia gandhii* Kanesharatnam & Benjamin (Salticidae) and *Pimoa gandhii* Hormiga (Pimoidae), dedicated to the Nobel Peace Prizes Nelson Mandela and Mohandas Karamchand Ghandi.

(b) *Largest colony*—Anelosimus eximius *(Keyserling) (Theridiidae)*. Among social spiders (see "*Most peaceful*"), *Anelosimus eximius* forms the largest cooperative groups (*GWR, 2017*). This species is found in rainforest in Central and South America. Communal webs may range in length from 10 to 25 cm containing only few individuals, to 2–3 m or more containing up to thousands of individuals (*Smith, 1986*). According to press media release ("Meet the spiders that have formed armies 50,000 strong" BBC—earth. Online at: www.bbc.com), some of the colonies may reach more than 7 m, containing as many as 50,000 individual spiders. However, it has been suggested that natural selection should actually favor intermediate rather than large colony sizes (*Avilés & Tufino, 1998*).

(c) *Best thieves*—Kleptoparasites. In spiders, best thieves are kleptoparasites, i.e., spiders regularly stealing food from the web of other spider species. Kleptoparasites generally

do not build webs, but exploit other spiders' webs for any of their activity. To date, kleptoparasitism has been documented in six spider families—Theridiidae, Dictynidae, Salticidae, Symphytognathidae, Mysmenidae and Mimetidae (*Vollrath, 1987*) (Fig. 4H).

(d) *Best camouflage—Shared by many species.* In the course of their evolution, many spider species have developed mimicry impressively (*Pekar, 2014*), and thus it is almost impossible to decide upon the best mimetic species. Spiders are able to mimic inanimate objects (masquerading mimicry), unpalatable or undesirable food in the eyes of their predators (Batesian mimicry), some of the habitat features in which they dwell (crypsis) or even specific pheromones produced by their prey (see "*Most creative hunting strategies*"). Examples of astonishing mimicries are found in spiders resembling bird dropping [e.g., *Cyclosa ginnaga* Yaginuma (Araenidae) (see, e.g., *Liu et al., 2014*)], ants [e.g., numerous species of Salticids and Thomisids (*Cushing, 2012*)], toads [e.g., *Poecilopachys australasia* (Griffith & Pidgeon) (Araneidae) (*Vink, 2015*)], seeds and fallen flowers [e.g., *Arachnura* spp. (Araneidae)] and leaves [e.g., *Poltys* sp. (Araneidae) (*Kuntner et al., 2016*)]. A very peculiar case of self-mimicry is given by *Cyclosa mulmeinensis* (Araneidae), which confound potential predators and parasitoids by attaching web decorations made by prey pellets that mimic its own body shape (*Tseng & Tso, 2009*).

(e) *Longest time under water—>16 h.* With the exception of *Argyroneta aquatica,* which conduct a wholly aquatic life (see "*Strangest habitat*"), there are other species that are able to conduct a partially aquatic life in intertidal habitats (see, also, "*Longest time under water in a nest*"). Using certain species of wolf spiders (Lycosidae), *Pétillon, Montaigne & Renault (2009)* compared survival rate during both a submersion and a recovery period after submersion. They found that salt-marsh species *Arctosa fulvolineata* (Lucas) (Lycosidae) is able to survive for more than 16 h underwater (100% mortality obtained at 36 h). This extraordinary survival was possible due to the spider ability to fall into a hypoxic coma, a physiological adaptation to overcome tidal inundation under water.

(f) *Longest time under water in a nest—Up to 19 days. Desis marina* (Hector) (Desidae) inhabits intertidal rocky habitats in New Caledonia and New Zealand (*WSC, 2017*). In these habitats, the species sometimes needs to survive up to 19 days of tide-induced submergence (*McQueen & McLay, 1983*). Despite lacking specific respiratory adaptations, *Desis marina* is able to hide away inside bull kelp holdfasts or sea worm burrows on the shore, blocking the water out with a lid woven of silk (*Rovner, 1986*; *GWR, 2017*).

## Ecology
### Habitat

(a) *Highest altitude—>6,000 m. Euophrys omnisuperstes* Wanless (Salticidae) owns the record of the spider dwelling at the highest altitude. A male specimen was collected at an altitude of around 5,900 m a.s.l. during an expedition in Mount Makalu (Nepal/

China). Immature specimens collected by Major Kingston at an altitude of around 6,700 m in Mount Everest (Nepal/China) were tentatively attributed to the same species (*Wanless, 1975*).

(b) *Lowest altitude—418 m below sea level.* The Dead Sea (Palestine, Israel and Jordan), is the lowest point on dry land—418 m below sea level. In their checklist of spiders from Israel *Zonstein & Marusik (2013)* reported 39 species occurring in this area, with representatives of the families Agelenidae, Araneidae, Cithaeronidae, Filistatidae, Gnaphosidae, Lycosidae, Oxyopidae, Prodidomidae, Salticidae, Scytodidae, Theridiidae and Thomisidae. Seven species have their own type locality (i.e., the locality where the species has been described) on the shore of the Dead Sea: *Pterotricha engediensis* Levy and *Talanites fervidus* Simon (Gnaphosidae), *Halodromus patellidens* (Levy) (Philodromidae), *Enoplognatha deserta* Levy & Amitai and *Theridion vallisalinarum* Levy & Amitai (Theridiidae), *Ozyptila rigida* (O. Pickard-Cambridge) (Thomisidae), and *Pax engediensis* Levy (Zodariidae).

(c) *Coldest place inhabited by spiders—Oymyakon (Russia).* The town of Oymyakon in Eastern Yakutia (Russia, 64.5° N—142.5° E) is known to be the coldest inhabited area of the planet, with an absolute minimal temperature of –71.2 °C recorded in 1924. *Marusik, Koponen & Potapova (2008)* documented the presence of 55 spider species living in Oymyakon and surroundings, including representatives of 11 families, mostly boreal or hypoarctic species of Gnaphosidae, Lycosidae and Linyphiidae.

(d) *Hottest place inhabited by spiders—Death Valley, Inyo, California.* The highest ground temperature on Earth (56.7 °C) was measured at Furnace Creek, Death Valley (CA) (*Kubecka, 2001*). A few authors (*Crews & Hedin, 2006*; *Crews & Gillespie, 2014*) reported about spiders living in this area, including wolf spiders (Lycosidae), mesh-web weavers (Dictynidae), jumping spiders (Salticidae), crab spiders (Thomisidae), cellar spiders (Pholcidae) and sand spiders (Homalonychidae). A few of them are exclusively found in very hot and salty areas and are considered true extremophiles. These are the wolf spider *Pardosa saltona* Dondale & Redner, the jumping spider *Habronattus tarsalis* (Banks), and the salt flat specialist *Saltonia incerta* (Banks) (S. Crews, 2017, personal communication).

(e) *Northernmost species—Erigone psychrophila Thorell (Linyphiidae).* Arctic spiders are reported from very high latitudes, including: Devon and Ellesmere Island (northern Canada) at 74–76° N; Greenland at 70–84° N; Iceland at 64–66° N; Jan Mayen Land at 71° N; Bear Island at 79° N; Svalbaard at 78° N; Novaya Zemlya at 71° N; Franz Josef Archipelago at 80–81° N (*Pugh, 2004*). Among these, the northernmost documented record is possibly the linyphiid *Erigone psycrophila.* It was collected during a scientific expedition by Mr. Henry Fisher at Cape Flora, Northbrook Island (Franz Joseph Arcipelago, Barents Sea) at 80° N, and later published by *Pickard-Cambridge (1898)*.

(f) *Southernmost species—Alien species in continental Antarctic.* Native spiders are absent from both Continental and Maritime Antarctica, the southernmost regions of the world. However, spider records from continental Antarctic exist, all representing dead,

anthropogenically imported, exotic or "alien" contaminants. These Antarctic aliens are *Erigone autumnalis* Emerton (Linyphiidae), one unidentified "Micryphantidae" (Linyphiidae) and an unidentified "Attidae" (Salticidae) from the Ross Sea coast of Continental Antarctica. Instead, the southernmost live records are from South Georgia (54° S; eight species, three of them alien) and Macquarie Island (54° S; seven species, two of them aliens), yet Neotropical species occur slightly further south at Tierra del Fuego (54–56° S) (*Pugh, 2004*).

(g) *Most diverse habitat—Atlantic forest.* The highest continental biodiversity on Earth is found in Brazil (*Brandon et al., 2005*), the largest tropical country in the world. This country present a variety of terrestrial ecosystems, including one of the major global biodiversity hotspot, the Atlantic Forest (*Myers et al., 2000*). For this biome, *Oliveira, Brescovit & Santos (2017)* provided the observed value of 1,672 species of spiders and estimated the impressive number of 2,714–3,816 species living therein. The highest species richness in the Atlantic Forest has also been reported for other taxa, such as flowering plants (*Sobral & Stehmann, 2009*), flatworms (*Carbayo et al., 2008*), dragonflies (*De Marco & Vianna, 2005*), and springtails (*Culik & Zeppelini-Filho, 2003*).

(h) *Least suitable habitat—Marine water.* No spiders evolved the ability to inhabit permanently submerged marine habitats, making it the most unsuitable habitat for spiders (but see "*Longest time underwater*" and "*Longest time underwater in a nest*"). The so-called sea spiders (Pycnogonida) are a very remote ancestor group of the Arachnids, but despite the name, it is wrong to consider them spiders. Although being traditionally classified as chelicerates, some features of this peculiar marine group suggest that they may be representatives of the earliest arthropods from which the Arachnids evolved (*Dunlop, 2010*).

(i) *Strangest habitat—Underwater.* Spiders are well-known to be ubiquitous in terrestrial ecosystem (*Foelix, 2011*). Being the only known spiders living a wholly aquatic life, we consider the diving bell spider *Argyroneta aquatica* (Clerck) (Cybaeidae) the species inhabiting the most peculiar habitat. *Argyroneta aquatica* has specific adaptations to breathe in immersion, being therefore able to hunt, to consume prey, to molt, to deposit eggs and to copulate underwater (*Seymour & Hetz, 2011*; *Mammola, Cavalcante & Isaia, 2016*) (Fig. 4F).

### Conservation

(a) *Rarest—Unclear.* In lack of detailed information about biology, ecology, range of distribution, and population size of the different species, rarity is extremely difficult to define from a biological viewpoint (*Gaston, 1994*). It is therefore challenging to assess which is the rarest spider species in the world. For instance, the *GWR (2017)* propose as the rarest spider the Kauai cave wolf spider [*Adelocosa anops* Gertsch (Lycosidae)], occurring in a few caves in the island of Kauai (covering a surface of circa 10.5 km$^2$). *Smithers & Whitehouse (2016)* suggested *Nothophantes horridus* Merrett & Stivens

(Linyphiidae) as the rarest spider in the world, being recorded exclusively from two abandoned limestone quarries near Plymouth, covering a surface of circa 0.1 km² (*Cardoso & Hilton-Taylor, 2015*). However, the reputation of 'rarest spiders' is possibly shared by numerous spiders described on the base of a single specimen, and never recorded thereafter (see *WSC, 2017*).

(b) *Most endangered—Shared by 36 species.* Thirty-six species of spiders are listed in the "critically endangered" IUCN category (*IUCN, 2015*), being therefore the most endangered species of spiders. Habitat changes and deterioration represent the major threats for these species. Some endangered Theraphosidae are also frequently commercialized as pets (see "*Most wanted as pet*"). However, it is worth noticing that only a minor part of the extant spider species has been evaluated against IUCN criteria (*Cardoso et al., 2011*)—currently 199 out of ~47,000 extant species (*IUCN, 2015*).

(c) *Most wanted as pet—Tarantulas.* As far as we are aware, the Gooty sapphire *Poecilotheria metallica* Pocock (Theraphosidae) is among the most commercialized spider species. According to *IUCN (2015)*, *Poecilotheria metallica* is considered "critically endangered," not only for the degradation of its natural habitat, but also due to its indiscriminate collection by pet traders. Since 2002, reports of advertised *Poecilotheria metallica* exported illegally from India and put on sale on the internet have been documented (*Molur, Daniel & Siliwal, 2008*).

## Curiosities

(a) *The longest journey—Into space.* In 1973, two females of *Araneus diadematus* Clerck (Araneidae) were sent into space on the Skylab 3 mission to the US Skylab space station (*Witt et al., 1976*). They are the first spiders that travelled in space (*GWR, 2017*). *Witt et al. (1976)* observed that web spun in space had modified features such as unusual distribution of radial angles and low number of turning points, which were attributed to the effect of the absence of gravity.

(b) *Most delicious—Personal preference.* It is difficult to assess which is the most delicious species of spider, as flavor is rather subjective and a matter of gourmets (see also "*Most eaten by humans*"). It is worth noting, however, that in some countries, spiders are considered a food delicacy. As an example, in Cambodia and Thailand *Haplopelma albostriatum* (Simon) (Theraphosidae) is served fried—but also canned with salt—as street food (*Ray, 2002*). A few species in Thailand are also used to flavor vodka and whiskey. In Venezuela, the jungle tribe Piaroa commonly eat *Theraphosa blondi* roasted.

(c) *Most eaten by humans—Many.* Most likely, the most eaten spiders are eaten accidentally. In many countries, the legal limits governing the presence of arthropods in processed foods are indeed large enough so that over time a large amount of spider parts is ingested (see, e.g., *CFSAN, 1998*).

(d) *Most feared—Indiscriminate.* Countless species of spiders terrify the public alike. With a prevalence rate ranging from 3.5% to 6.1% of the population (*Jacobi et al., 2004*;

*Schmitt & Müri, 2009*), "arachnophobia" is indeed documented to be the most common phobia related to animals (*Hofmann, Alpers & Pauli, 2009*).

(e) *Largest item of clothing woven from spider silk—A lady's cape.* The American fashion designer Nicholas Godley and textile expert Simon Peers masterminded and created the largest item of clothing woven from spider silk: a lady's cape with matching 4-m long brocade scarf containing ca. 1.5 kg of silk. The silk used was woven by more than one million females of *Nephila* (Araneidae) (*GWR, 2017*).

(f) *Most iconic spider—Spiderman.* Arachnid symbolism is found through human history (*Melic, 2002*). Possibly, the most famous, successful and iconic character inspired by arachnids is Spiderman, the famous Marvel superhero created by Stan Lee and Steve Ditko in 1962—see the official GWR for a number of records related to Spiderman. However, it is worth noticing that, according to a recent survey (*Da-Silva et al., 2014*), Arachnids inspired at least 123 other comics characters in the comics literature.

## DISCUSSION

Spiders have a bad reputation among the general public (*Jacobi et al., 2004*; *Schmitt & Müri, 2009*): they are considered ugly, hairy, brown, and deadly poisonous creatures. There are tales describing how they lay eggs in human skin, frequent toilet seats in airports, and crawl into your mouth when you are sleeping. Misinformation about spiders in the popular media and on the World Wide Web is often rampant, leading to distorted perceptions and negative feelings about spiders. However, despite their negative connotation, spiders offer intrigue and mystery and can be used to effectively engage even arachnophobic people into arachnid-based discussions and activities. Toward this end, this original list of record breaking spider achievements provides a wide range of entry points into the rich biology of spiders. The numerous facts, observations, and even unknowns compiled herein (99 records) offer intriguing content and inspiration for educators, provide engaging hooks for students and learners of all ages, and highlight potentially fruitful new directions for future scientific research. Given the scarcity of database such as this, our work can provide a framework and foundation to which others can contribute.

For the scholars among us, whose interests encompass the history of science, we reveal in our section on *Arachnology and arachnologists* that scientists have been interested in spiders since the early 1700s. In reading these early published works, their predominantly descriptive nature and focus on natural history is notable and is found to contrast strongly with the style of current primary scientific publications. Despite the shift of focus and style, however, scientists today remain fascinated by spiders. Fortunately, the number of arachnologists and the diversity of arachnological studies do not appear to be diminishing. The largest congress of arachnologist in history, for example, was as recent as 2016. We expect that arachnology will remain strong and hope that this contribution will help to draw future arachnologists into the world of spider research.

In the section on *Paleontology*, we anticipate that the extensive evolutionary history of spiders is also notable to educators. In contrast to the vertebrate groups, that are often at

the forefront of one's mind when discussing "animals" [e.g., mammals, Late triassic, 237–201 Myr ago (*Benton, 2005*); birds, ca. 70 Myr ago (*Prum et al., 2015*)], spiders have inhabited our planet for at least 300 million years. Interesting, however, we show that the earliest recorded spider silk dates back to ~140 Myr ago. Readers might wonder—why is there such a discrepancy between the timing of spider fossils and silk records? We suggest that such a question could facilitate further research into the process of fossilization and the preservation of different biological materials. Additionally, armed with the knowledge that spiders have multiple silk glands and can produce different types of silk with distinct physical properties, readers might now wonder — what type(s) of silk was present 140 Myr ago? Did spiders always build webs, or did webs evolve more recently? Again, such questions could motivate further research among interested students. From this additional research, they could learn that the earliest spiders did not build webs and, in fact, the vertical orientation of the orb webs did not evolve until insects took the air in flight (*Bond & Opell, 1998*). Thus, the history of spider silk use provides an appealing and accessible storyline for teaching about evolutionary change. Indeed, one arachnid-based informal science event, that has successfully travelled to multiple venues across the United States of America, incorporates silk-related games and activities to demonstrate both the diversity and evolutionary history of spider silk form and function (*Eight-Legged Encounters*; http://hebetslab.unl.edu/eight-legged-encounters/spiders-and-silk/).

The *Taxonomy and Systematics* section provides baseline information and facts regarding the biodiversity of spiders. From reading this section, one might wonder why jumping spiders, in particular, are the most diverse spider family. In another section (*Physiology—Sensory organs*) readers learn that jumping spiders also have the best diurnal eyesight among all spiders. They are also cited as having the most elaborate courtship (*Behavior—Reproduction*). Is there a relationship then between visual capacity, courtship behavior, and diversification? Curiously, the other spider family with good diurnal eyesight—wolf spiders (Lycosidae)—are also fairly diverse (>2,000 species; *WSC, 2017*) and some genera within this family are also known for their complex courtship displays [e.g., *Schizocosa* (*Hebets et al., 2013*); *Pardosa* (*Chiarle & Isaia, 2013*)]. Research attempting to understand the potential relationship(s) between diversification (i.e., species number), visual capacity, and reproductive behavior could provide important insights into our understanding of speciation—e.g., the putative role of sensory physiology.

Our section on *Anatomy* follows the basic spider body structure that we presented in the section *Brief Introduction to Spiders*. The first prominent records highlight the incredible size range of spiders, with the largest spiders measuring almost 40 mm in length and the smallest less than 0.4 mm. This size range represents a 100-fold difference between the largest and smallest spiders. Do these spiders have similar lifespans? Do they go through a similar number of molts? If so, are there fundamental differences in their metabolic rates or other aspects of their physiology that can account for observable differences in growth and development? To the best of our knowledge, these are still open questions.

In our *Anatomy* section we also present records associated with measurable body parts and appendages—e.g., chelicerae, walking legs. Though the records in this section should

be straightforward and uncontroversial, we found them to be difficult to ascertain in many instances. For example, to be informative and useful toward our goals, structural records need to be related to overall body size—e.g., largest legs relative to body size. Not only are most published size measurements not calculated in relation to body size, but published numbers also tend to be buried in very old species descriptions. We maintain, however, that such information on anatomical relative size could be incredibly informative for both teaching evolutionary concepts, and for guiding future research efforts.

Species with particularly long fangs, for example, likely have a unique foraging strategy or prey type—e.g., the unusual shapes of the chelicerae and fangs of spiders in the family Dysderidae are often specializations for feeding on woodlice (*Cooke, 1965*; *Řezáč & Pekar, 2007*). Unusually large or atypically shaped spinnerets may indicate something original about the way in which silk is laid or produced, or may reflect novel aspects of the silk itself. As such, spiders with unusual spinnerets may be fruitful focal taxa for studies of web structure and design or silk production and composition. Similarly, species with particularly long legs relative to their body may provide good focal taxa for exploring mechanisms underlying locomotion, the findings of which could potentially stimulate new designs in robotics. We encourage arachnologists to examine our current external anatomy records for peculiarities that might deserve further focused attention, but also to be diligent about incorporating basic measurement information in future publications, such that new records can be readily found and documented.

Our *Internal organs* subsection (*Anatomy*) is admittedly the sparsest and incorporates the most speculation. This is due to the fact that documentation and assessment of variation in internal anatomical structures is not typical of scientific studies, unless there is a very specific research question associated with the data collection. Regardless, this section remains important as it highlights additional area(s) where opportunities for discovery may exist. Which is the spider with the largest relative heart? Why? Unusually large, or small, hearts could suggest physiological challenges and/or adaptations related to respiration and circulation. A priori, it is impossible to foresee how knowledge of such adaptations might be useful or informative—e.g., for innovation related to human health. Some of our documented records in this section highlight the potential importance of internal anatomical records. For example, we include records demonstrating that small spiders have proportionally large brains that take up an impressive portion of their body cavity (*Quesada et al., 2011*). This observation raises fundamental questions about the constraints that small animals may face in terms of brain size and associated behavior. This record breaking achievement can also be used to guide students through fundamental information regarding cell biology and nervous system form and function. It can, for example, guide students through asking and answering fundamental questions such as: How variable in size are animal cells? Why? Are all axons within and/or among animals of similar diameter? Is there an upper or lower limit to axon diameter? Ultimately, while we have certainly provided a starting point for internal anatomical records, we urge scientists to pay closer attention to variation in internal anatomy both within and among spider species, as we see this as a particularly fruitful area of future research inspiration and discovery.

Many animal physiologists adopt the Krogh principle (*Krogh, 1929*), which states that "*for such a large number of problems there will be some animal of choice, or a few such animals, on which it can be most conveniently studied.*" We expand on this principle by proposing that the problems do not need to exist a priori, but instead animals themselves can present problems or puzzles for us to study. For example, in our *Physiology* section, we highlight the shortest circadian rhythm recorded. This new research raises a number of questions. What is the circadian rhythm of most spiders? How and why might circadian rhythms vary within and across taxonomic groups? Similarly, it is in this section that we highlight new research documenting the capacity of a jumping spider to perceive airborne sound. Though there is evidence of hearing in this jumping spider (*Shamble et al., 2016*), the mechanism underlying this capacity remains enigmatic, thus opening up new avenues for future research. Many of our other documented record breaking achievements can guide students through a range of questions: how do animals tolerate extreme environments? Why doesn't the blood of spiders freeze in the winter? Or, how might fundamental knowledge of animal sensory systems inspire technological innovation—e.g., the development of new microphones based on the biology of spider vibratory senses (*Kang et al., 2014*).

Our compilation of behavioral and ecological record breaking achievements were two of the easier sections to pull together and can likely be expanded upon greatly in the future. The behavioral diversity of spiders has been leveraged by ethologists for centuries, and syntheses and compilations of this rich repertoire already exist (*Herberstein, 2011*). Due to their range of reproductive behavior and mating systems, foraging strategies, communication systems, and lifestyles (among others), spiders provide excellent models for teaching and learning about behavioral evolution (*Herberstein & Hebets, 2013*). Their ethology has already facilitated research on a range of topics from sexual selection (*Huber, 2005*) to sperm dynamics (*Herberstein et al., 2011*), and there are seemingly unlimited possibilities for the future.

Furthermore, we pointed out the extreme ecological plasticity of this successful group of Arthropods. They reach more than 6,000 m altitude, they survive in the hottest and coldest places on Earth, they colonize almost all types of ecosystems—one exception, marine underwater—and exhibit extraordinary values of diversity, especially in the Tropics. Furthermore, being mostly predators (but see "*Strangest diet*") they play a fundamental role in the ecosystem. Despite their ecological importance, the conservation issues about this animal group is largely neglected (*Rix et al., 2016*). In fact, the extinction risk of a very minimal portion of the known spider diversity has been assessed (see "*Most endangered*"). However, global threats such as habitat loss, fragmentation and climate change are likely to affect the survival of a vast number species inhabiting a range of different habitats (*Leroy et al., 2013*, *2014*; *Kuntner et al., 2014*; *Mammola, Goodacre & Isaia, 2017*).

It is notable that many of our incorporated records have been published since 2010. This accurately reflects the relative infancy of arachnology relative to other organismal systems such as mammals, birds, or even insects. By some estimates, arachnologists have described only one third of the spider species worldwide (*Agnarsson, Coddington &*

*Kuntner, 2013*); and even among the described species, basic information about their biology and natural history remain unknown. Indeed, our knowledge of spiders is still in its early stages and, with the expected future discoveries of thousands of new species and novel observations of species already known to science, will surely come new records and new curiosities. We also acknowledge that our list of record breaking achievements is far from exhaustive and it is certainly possible that records hidden in old publications or written in inaccessible languages (to us) may have been missed.

In summary, with their incredible diversity, spiders provide outstanding examples of how increased knowledge, understanding, and appreciation of a specific group of organisms can facilitate learning and understanding of science and nature, increase the public's enthusiasm for and connection with the natural world, and simultaneously push the envelope of science forward in a number of distinct directions. We hope that this compilation of record breaking achievements helps spiders to achieve their teaching, learning, and research potential. We also documented some discrepancies between the information found in the scientific literature and those in the official GWR database (*GWR, 2017*), thus we are able to provide suggestions for updates and corrections (see Supplemental information). Finally, in order to transform this database into a community-driven knowledge base, we will implement these records on the website of the International Society of Arachnology (www.arachnology.org). We very much see this as a living document that will grow and change as new knowledge is gained and new discoveries are made.

## ACKNOWLEDGEMENTS

Many thanks to all friends and colleagues who posed to us bizarre questions about spiders, stimulating the idea for this paper. Rebecca Wilson, Yael Lubin, Yuri Marusik, Theo Blick, Jens Runge, Philippe Vernon, Filippo Milano, Alexandra Jones, Sarah Crews, Raquel Galindo, Colton Watts, Rowan McGinley, Noori Choi, Alissa Anderson, Cecilia Ruffino, and Silvia Grilli provided information and/or suggested some of the records listed herein. We thank Irene Frigo for the help in creating the layout of Fig. 1. We are grateful to all photographers for sharing their photos of spiders (see captions of Figs. 2–4). Special thanks is due to Sergio Henriques and two anonymous referees for useful comments and suggestions to improve the manuscript.

### Funding

The authors received no funding for this work.

### Competing Interests

The authors declare that they have no competing interests.

### Author Contributions

- Stefano Mammola conceived and designed the experiments, wrote the paper, prepared figures and/or tables, reviewed drafts of the paper, assembled the first draft of records.

- Peter Michalik wrote the paper, reviewed drafts of the paper, suggested additional records, is responsible for the online publication of the records.
- Eileen A. Hebets wrote the paper, reviewed drafts of the paper, suggested additional records, gave the educational focus to the paper.
- Marco Isaia wrote the paper, prepared figures and/or tables, reviewed drafts of the paper, suggested additional records, supervised this study.

## Data Availability

The record list will be linked to the website of the International Society of Arachnology (www.arachnology.org). This research has no raw data or codes (literature review).

## Supplemental Information

Supplemental information for this article can be found online at http://dx.doi.org/10.7717/peerj.3972#supplemental-information.

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
