# Peer review of "Record breaking achievements by spiders and the scientists who study them"

_PeerJ, doi:10.7717/peerj.3972_

## Round 0.1 · original submission · Major Revisions

· Academic Editor

Major Revisions

Three expert reviewers evaluated your manuscript. Two of them suggested "Major revisions", while one suggested "Reject". From my own reading, I see merit in your manuscript. However, a considerable reworking is needed. Therefore, I offer you the possibility to address the reviewers' comments carefully in a revision round.

·

Basic reporting

Overall, the paper is clearly innovative and has the potential to provide valuable contribution to the field of arachnology.

However I believe that its main focus can become ambiguous, since some of central terminology used can be misinterpreted. Particularly the term "record", which in arachnological research is often linked to geographical occurrences of species.
(eg.
Estol, N. and RODRIGUES, E.N.L., 2017. A new species and a new RECORD of the spider genus Nesticus (Araneae, Nesticidae) from southern Brazil. Zootaxa, 4231(4), pp.564-566.

Malamel, J.J. and Ambalaparambil, S., 2017. First RECORD of Epidius parvati Benjamin, 2000 (Araneae: Thomisidae) from Pathiramanal Island, India. Check List, 13(3), p.2114.

DEMİR, H., SEYYAR, O. and NAJİM, S.A., 2017. Thomisus citrinellus Simon 1875 is a new RECORD (Araneae: Thomisidae) for spider fauna of Iraq.
)

This does not mean the title should loose it's outreach message, or its focus towards reaching and engaging with a broader audience. In fact the title can be rewritten to reinforce this objective, while making it clear what "record" means in this context (eg. Record breaking achievements by some of the world's smallest champions: A scientific frame work for the first database of World Record Holding Spiders).

The title section "a resource for using organismal biology as a hook for science learning" is neither intuitive to understand (e.g. organismal biology), nor a part of the publication analysis.

I don't disagree with the statement, and potential areas towards which the publication can be impactful should be mentioned, but since they were not a part of the analysis, it is my personal opinion that they should not be referred in the title.

Literature references provide a good background in the context of this research, but were neither exhaustive, not systematic. It is perhaps impossible, or unfeasible to produce on a comprehensive scale, but it would be clearly valuable to produce a systematic one (I mention this approach in more detail at the "validity of the findings" section).

Experimental design

The primary research fits well within the area of "Literature Review Articles"and this was how I accessed it's scientific and methodological merit.

The work is relevant and meaningful, but I don't believe the research question was well defined, and it doesn't state clearly how it fills and identifies knowledge gaps. I believe this is mostly due to to a lack of a rigorous approach, and a lack of information on how to replicate the study.

Although this was not (neither can it realistically be) a comprehensive review of spider record holding achievements published in the literature, it is my opinion that the authors should:
a) have built a structured scientific framework for the set of traits that can be measured or analysed in a standard way, and b) attempt to produce a comprehensive or representative systematic bibliographic review. An approach which would not only enhance the manuscript but also allow for it's reproducibility.

Therefore, I would strongly encourage the authors to use their "Brief Introduction to Spiders", as base point to define which metrics can be considered for record achievement (including the ones which were considered but no scientific data was found). Generating a framework that methodically revises the current knowledge and pinpoints unstudied fields of research, allowing other arachnologists to built upon it.

In practice, I am suggesting revising the proposed "four general categories" in a methodical manner.
(E.g.
Category II - Morphology
Section a - Prossoma
Characters 1 - external;
Traits: Overall traits (biggest/smallest; proportionally larger/smaller; flattest/tallest carapace, most/least colourful...)
Eyes (largest/smallest; proportionally larger/smaller...; for best eyesight see cat.III physiology)
Chelicerae/fangs (longest/shortest; proportionally larger/smaller; wider/thinner...; for venom toxicity or wider opening angle see cat.III physiology; for quicker closing time see cat. IV behaviour)
Ornaments (hairiest/baldest; most cuticular spines - Aphantauchilus...

Characters 2 - internal;
Traits: Venom gland (smallest/biggest; proportionally larger/smaller; etc...; for venom toxicity see cat.III physiology)
Brain (smallest/biggest; proportionally larger/smaller; ... for best memory see Portia fimbriata cat IV behaviour)

Section b - Opistossoma
Characters 1 - external;
Traits: etc...
)

This sort of methodically approach (eg. comprehensively listing all morphological traits that can be feasibly measured) would facilitate the analysis of the database, and the detection of unstudied/unrecorded areas, and when broadly applied to all the categories explored would not only include more physiological traits (eg. longest/largest/smallest male/female reproductive organs), but also:

- more environmental features (e.g. the hottest/coldest habitat dweller);
- more behavioural traits (eg. the loudest spider, see Allard, 1936, Davis, 1904; Edwards, 1981, Lahee, 1904; Prell, 1916; Rovner, 1980; Uetz& Stratton, 1982);
- physiological achievements (eg. longest period under freezing temperatures, longest time under water, see Pétillon et al. 2009)
- more behavioural achievements (eg. organic projectiles other than silk, since spiders are able to project venom - Scytodes, or excrete liquid from their anus - Hummidia, to considerable distances);
- other traits (eg. best body mimic: of leafs -Arachnura, of toads - Poecilopachys australasia; of ants - Aphantauchilus; most realist body double - Cyclosa sp.; etc...)

All of which clearly fit the publication premise but were not assessed, potentially because these traits were not researched methodically, but rather collected out of availability.


No such list can ever be truly comprehensive, but I believe this approach, even if it generates multiple unknown record holding positions, would clearly allow for gaps to be made clearer, and incentivise other to either provide information about those gaps, or research those fields.

Another way to tackle large data gaps, might also be the analysis of local record holders (eg. the most venomous spider in Europe, or the most venomous in the US).


Besides producing a rigorous analysis of traits, I would also suggest the authors to perform a systematic bibliographic review on all, or at least on one of the category keywords (as a sampled approach). Using a reference-based search strategy, which would be both reproducible and insure that all the bibliography currently available (in the selected database) was comprehensively revised.

As a guideline I recommend analysing the methods used in:

Savilaakso, S., Garcia, C., Garcia-Ulloa, J., Ghazoul, J., Groom, M., Guariguata, M.R., Laumonier, Y., Nasi, R., Petrokofsky, G., Snaddon, J. and Zrust, M., 2014. Systematic review of effects on biodiversity from oil palm production. Environmental Evidence, 3(1), p.4.

Reed, J., van Vianen, J., Foli, S., Clendenning, J., Yang, K., MacDonald, M., Petrokofsky, G., Padoch, C. and Sunderland, T., 2017. Trees for life: the ecosystem service contribution of trees to food production and livelihoods in the tropics. Forest Policy and Economics.

Validity of the findings

As it stands the results can be considered inconclusive, however the suggestions made above should allow the publication to obtain robust data that provides statistically sound conclusions on which specific areas or research fields have been the most, and the least investigated in arachnological literature.

Reviewer 2 ·

Basic reporting

This paper does not necessarily take a form of hypothesis-test setting. Various interesting records about spider are presented, but these records are not related to each other, and are not well integrated in discussion. Therefore, I feel that this paper was somewhat descriptive and distracting. In review thesis, the authors should find new perspectives from heterogeneous findings and phenomena.

Experimental design

No comment.

Validity of the findings

This is a very useful review that allows ordinary people to access interesting records of spiders. However, as it is a list of known facts, it seemed that there is lack of heuristic elements. I felt that it is necessary to find synergistic findings obtained by gathering various records of spiders. As the authors say, it would be better to provide examples of creative discussion (L.708-709).

Additional comments

This paper is an article that summarizes interesting records of spiders with World Record format, based on academically correct information to remove the misunderstanding concerning spiders. Because it is an unprecedented type of paper which is different from the typical review article, it is very difficult to evaluate, but I felt that it could be useful literature to inform the spider world. On the other hand, from the viewpoint of academic novelty, I felt this paper is somewhat descriptive, and the new perspective obtained by gathering many records is unclear. Also, from the viewpoint of generality, since this paper only focuses on the spider topic, I got a somewhat specific impression. In order to emphasize “spider’s greatness”, I think that it would be better to have a comparison with world records of other taxonomic group such as insects. I think that this paper does not necessarily need to be published in the general science journal, and recommend submission in another journal dealing more specifically with entomological or arachnological issues.

Reviewer 3 ·

Basic reporting

Review of Peer J #18552
This is an interesting paper that brings together information about spider biology and taxonomy with the intend to inspire the public and educators to engage with the natural world. I agree that spiders have a privileged position as most people have some opinion about spiders and often react positively when provided with more information.
Overall, I believe this paper will make an interesting contribution and only have few comments to make about its readability.


Writing style – overall the paper is very well written and entertaining to read. My comments are:
The terminology used throughout the paper is confusing and makes it unclear who the intended readership is. Is it biologists? Is it lay-people? If it is the later, then terms need to be explained.

Throughout (e.g. 60, 62) phrases are placed in parentheses, but they do not have to.
75-76: sentence too long and convoluted – please rephrase
Overall the introduction is quite lengthy and could be shortened and sharpened. Eg. Is 87-92 necessary?
106: Here we build on records from the scientific…
140: The body of a spider is divided….
140 and onwards – please refer to Fig 1
146 & 147 & 149: ‘behind’…some of the terminology used in this paper is quite sophisticated, so I do not see why the term ‘posterior’ cannot be used instead of ‘behind’
152: from the ophistosoma
156: terminology already explained earlier
162: remove ‘so called’
163: explain terminology earlier
174: why what?
177: use the term ‘collaboratively’
178-181: repetitive
187-190: repetitive
223: First and last listed spider
238: what do the terms in the parentheses mean?
237-239: sentence complex and convoluted, please rephrase
244-245: please explain these obscure terms (nomen dubia; synonymized)
313: explain GB
340: explain holotype
375: the least number…
376: explain anophthalmic
384: capable of spinning…
386: ‘dimension’ – do you mean ‘diameter’?
394: explain MJ
397: explain GPa
418: why ‘sic’
419: explain hypogean
426: greatest longevity
482: in what year?
547: what do you mean by conscious?
581: a small number…
620: plant products
688: I found the discussion quite repetitive, with arguments that have already been presented in earlier sections.
Rather than repeat this information, maybe the discussion could provide some examples of how educators might utilise this information

Experimental design

no applicable

Validity of the findings

not applicable

Additional comments

The idea of this paper is great, I really like it and it almost works, but for the Discussion which needs some work.

---

## Round 0.2 · Minor Revisions

· Academic Editor

Minor Revisions

As you can see only very minor revisions are requested before acceptance.

Reviewer 2 ·

Basic reporting

The revision of the Discussion section by the authors seems to solve the problems I was concered about in the previous review. English expression and cited references are also appropriate.

Experimental design

no comment

Validity of the findings

I think that the purpose and perspective of this paper became clear by the revision of the Discussion section. No problem.

Additional comments

In the previous review, I was concerned about whether this MS fits the concept of the "PeerJ", but I changed my mind that there is no problem with this point after reading the author's response. In my country, there were no such literature which summarizes interesting records of spiders, so it seems very useful for widely communicating the attractiveness of spiders to people. Additionally, I think that this review is meaningful in presenting a new framework of the review paper. I judge that this paper is worth publishing.

Reviewer 3 ·

Basic reporting

Review Peerj-18552

I have seen the original version of this manuscript and the authors have addressed my concerns. In particular, the discussion is much improved. I only have picked up a few minor items.
Line 61-62: remove – unsubstantiated claim
333: the term ‘orb-web spider’ is more accurate. Spiders don’t weave
361: are fangs different from chelicerae? Please check terminology
402: so there are no species that have lost the venom gland? Can you clarify this point? See line 454
466: there are plenty of original papers that attest to the visual acuity of jumping spiders (e.g. work by Zurek)
499: eggs and sperm

Experimental design

no applicable

Validity of the findings

not applicable

Additional comments

Review Peerj-18552

I have seen the original version of this manuscript and the authors have addressed my concerns. In particular, the discussion is much improved. I only have picked up a few minor items.
Line 61-62: remove – unsubstantiated claim
333: the term ‘orb-web spider’ is more accurate. Spiders don’t weave
361: are fangs different from chelicerae? Please check terminology
402: so there are no species that have lost the venom gland? Can you clarify this point? See line 454
466: there are plenty of original papers that attest to the visual acuity of jumping spiders (e.g. work by Zurek)
499: eggs and sperm

---

## Round 0.3 · accepted · Accept

· Academic Editor

Accept

The manuscript has been properly revised, it can be accepted now.